# Reversible and irreversible bracket-based dynamics for deep graph neural networks

**Anthony Gruber** [†]
Center for Computing Research
Sandia National Laboratories
Albuquerque, NM. USA
adgrube@sandia.gov

**Kookjin Lee** [*]
School of Computing and Augmented Intelligence
Arizona State University
Tempe, AZ. USA
kookjin.lee@asu.edu

**Nathaniel Trask** [†]
School of Engineering and Applied Science
University of Pennsylvania
Philadelphia, PA. USA
ntrask@seas.upenn.edu

## Abstract

Recent works have shown that physics-inspired architectures allow the training of deep graph neural networks (GNNs) without oversmoothing. The role of these physics is unclear, however, with successful examples of both reversible (e.g., Hamiltonian) and irreversible (e.g., diffusion) phenomena producing comparable results despite diametrically opposed mechanisms, and further complications arising due to empirical departures from mathematical theory. This work presents a series of novel GNN architectures based upon structure-preserving bracket-based dynamical systems, which are provably guaranteed to either conserve energy or generate positive dissipation with increasing depth. It is shown that the theoretically principled framework employed here allows for inherently explainable constructions, which contextualize departures from theory in current architectures and better elucidate the roles of reversibility and irreversibility in network performance. Code is available at the Github repository https://github.com/natrask/BracketGraphs.

[*]K. Lee acknowledges the support from the U.S. National Science Foundation under grant CNS2210137.

[†]Sandia National Laboratories is a multimission laboratory managed and operated by National Technology & Engineering Solutions of Sandia, LLC, a wholly owned subsidiary of Honeywell International Inc., for the U.S. Department of Energy's National Nuclear Security Administration under contract DE-NA0003525. This paper describes objective technical results and analysis. Any subjective views or opinions that might be expressed in the paper do not necessarily represent the views of the U.S. Department of Energy or the United States Government. This article has been co-authored by an employee of National Technology & Engineering Solutions of Sandia, LLC under Contract No. DE-NA0003525 with the U.S. Department of Energy (DOE). The employee owns all right, title and interest in and to the article and is solely responsible for its contents. The United States Government retains and the publisher, by accepting the article for publication, acknowledges that the United States Government retains a non-exclusive, paid-up, irrevocable, world-wide license to publish or reproduce the published form of this article or allow others to do so, for United States Government purposes. The DOE will provide public access to these results of federally sponsored research in accordance with the DOE Public Access Plan https://www.energy.gov/downloads/doe-public-access-plan. The work of N. Trask and A. Gruber is supported by the U.S. Department of Energy, Office of Advanced Computing Research under the "Scalable and Efficient Algorithms - Causal Reasoning, Operators and Graphs" (SEA-CROGS) project, the DoE Early Career Research Program, and the John von Neumann fellowship at Sandia.

# 1    Introduction

Graph neural networks (GNNs) have emerged as a powerful learning paradigm able to treat un-structured data and extract "object-relation"/causal relationships while imparting inductive biases which preserve invariances through the underlying graph topology [1, 2, 3, 4]. This framework has proven effective for a wide range of both *graph analytics* and *data-driven physics modeling* problems. Despite successes, GNNs have generally struggle to achieve the improved performance with increasing depth typical of other architectures. Well-known pathologies, such as oversmoothing, oversquashing, bottlenecks, and exploding/vanishing gradients yield deep GNNs which are either unstable or lose performance as the number of layers increase [5, 6, 7, 8].

To combat this, a number of works build architectures which mimic physical processes to impart desirable numerical properties. For example, some works claim that posing message passing as either a diffusion process or reversible flow may promote stability or help retain information, respectively. These present opposite ends of a spectrum between irreversible and reversible processes, which either dissipate or retain information. It is unclear, however, what role (ir)reversibility plays [9]. One could argue that dissipation entropically destroys information and could promote oversmoothing, so should be avoided. Alternatively, in dynamical systems theory, dissipation is crucial to realize a low-dimensional attractor, and thus dissipation may play an important role in realizing dimensionality reduction. Moreover, recent work has shown that dissipative phenomena can actually sharpen information as well as smooth it [10], although this is not often noticed in practice since typical empirical tricks (batch norm, etc.) lead to a departure from the governing mathematical theory.

In physics, Poisson brackets and their metriplectic/port-Hamiltonian generalization to dissipative systems provide an abstract framework for studying conservation and entropy production in dynamical systems. In this work, we construct four novel architectures which span the (ir)reversibility spectrum, using geometric brackets as a means of parameterizing dynamics abstractly without empirically assuming a physical model. This relies on an application of the data-driven exterior calculus (DDEC) [11], which allows a reinterpretation of the message-passing and aggregation of graph attention networks [12] as the fluxes and conservation balances of physics simulators [13], providing a simple but powerful framework for mathematical analysis. In this context, we recast graph attention as an inner-product on graph features, inducing graph derivative "building-blocks" which may be used to build geometric brackets. In the process, we generalize classical graph attention [12] to higher-order clique cochains (e.g., labels on edges and loops). The four architectures proposed here scale with identical complexity to classical graph attention networks, and possess desirable properties that have proven elusive in current architectures. On the reversible and irreversible end of the spectrum we have *Hamiltonian* and *Gradient* networks. In the middle of the spectrum, *Double Bracket* and *Metriplectic* architectures combine both reversibility and irreversibility, dissipating energy to either the environment or an entropic variable, respectively, in a manner consistent with the second law of thermodynamics. We summarize these brackets in Table 1, providing a diagram of their architecture in Figure 1.

**Primary contributions:**

**Theoretical analysis of GAT in terms of exterior calculus.** Using DDEC we establish a unified framework for construction and analysis of message-passing graph attention networks, and provide an extensive introductory primer to the theory in the appendices. In this setting, we show that with our modified attention mechanism, GATs amount to a diffusion process for a special choice of activation and weights.

**Generalized attention mechanism.** Within this framework, we obtain a natural and flexible extension of graph attention from nodal features to higher order cliques (e.g. edge features). We show attention must have a symmetric numerator to be formally structure-preserving, and introduce a novel and flexible graph attention mechanism parameterized in terms of learnable inner products on nodes and edges.

**Novel structure-preserving extensions.** We develop four GNN architectures based upon bracket-based dynamical systems. In the metriplectic case, we obtain the first architecture with linear complexity in the size of the graph while previous works are $O(N^3)$.

**Unified evaluation of dissipation.** We use these architectures to systematically evaluate the role of (ir)reversibility in the performance of deep GNNs. We observe best performance for partially

| Formalism | Equation | Requirements | Completeness | Character |
|---|---|---|---|---|
| Hamiltonian | $\dot{\mathbf{x}} = \{\mathbf{x}, E\}$ | $\mathbf{L}^* = -\mathbf{L}$, Jacobi's identity | complete | conservative |
| Gradient | $\dot{\mathbf{x}} = -[\mathbf{x}, E]$ | $\mathbf{M}^* = \mathbf{M}$ | incomplete | totally dissipative |
| Double Bracket | $\dot{\mathbf{x}} = \{\mathbf{x}, E\} + \{\{\mathbf{x}, E\}\}$ | $\mathbf{L}^* = -\mathbf{L}$ | incomplete | partially dissipative |
| Metriplectic | $\dot{\mathbf{x}} = \{\mathbf{x}, E\} + [\mathbf{x}, S]$ | $\mathbf{L}^* = -\mathbf{L}, \mathbf{M}^* = \mathbf{M}$, $\mathbf{L}\nabla S = \mathbf{M}\nabla E = \mathbf{0}$ | complete | partially dissipative |

Table 1: The abstract bracket formulations employed in this work. Here $\mathbf{x}$ represents a state variable, while $E = E(\mathbf{x}), S = S(\mathbf{x})$ are energy and entropy functions. "Conservative" indicates purely reversible motion, "totally dissipative" indicates purely irreversible motion, and "partially dissipative" indicates motion which either dissipates $E$ (in the double bracket case) or generates $S$ (in the metriplectic case).

dissipative systems, indicating that a combination of both reversibility and irreversibility are important. Pure diffusion is the least performant across all benchmarks. For physics-based problems including optimal control, there is a distinct improvement. All models provide near state-of-the-art performance and marked improvements over black-box GAT/NODE networks.

## 2    Previous works

**Neural ODEs:** Many works use neural networks to fit dynamics of the form $\dot{x} = f(x, \theta)$ to time series data. Model calibration (e.g., UDE [14]), dictionary-based learning (e.g., SINDy [15]), and neural ordinary differential equations (e.g., NODE [16]) pose a spectrum of inductive biases requiring progressively less domain expertise. Structure-preservation provides a means of obtaining stable training without requiring domain knowledge, ideally achieving the flexibility of NODE with the robustness of UDE/SINDy. The current work learns dynamics on a graph while using a modern NODE library to exploit the improved accuracy of high-order integrators [17, 18, 19].

**Structure-preserving dense networks:** For dense networks, it is relatively straightforward to parameterize reversible dynamics, see for example: Hamiltonian neural networks [20, 21, 22, 23], Hamiltonian generative networks [24], Hamiltonian with Control (SymODEN) [25], Deep Lagrangian networks [26] and Lagrangian neural networks [27]. Structure-preserving extensions to dissipative systems are more challenging, particularly for *metriplectic* dynamics [28] which require a delicate degeneracy condition to preserve discrete notions of the first and second laws of thermodynamics. For dense networks such constructions are intensive, suffering from $O(N^3)$ complexity in the number of features [29, 30, 31]. In the graph setting we avoid this and achieve linear complexity by exploiting exact sequence structure. Alternative dissipative frameworks include Dissipative SymODEN [32] and port-Hamiltonian [33]. We choose to focus on metriplectic parameterizations due to their broad potential impact in data-driven physics modeling, and ability to naturally treat fluctuations in multiscale systems [34].

**Physics-informed vs structure-preserving:** "Physics-informed" learning imposes physics by penalty, adding a regularizer corresponding to a physics residual. The technique is simple to implement and has been successfully applied to solve a range of PDEs [35], discover data-driven models to complement first-principles simulators [36, 37, 38], learn metriplectic dynamics [39], and perform uncertainty quantification [40, 41]. Penalization poses a multiobjective optimization problem, however, with parameters weighting competing objectives inducing pathologies during training, often resulting in physics being imposed to a coarse tolerance and qualitatively poor predictions [42, 43]. In contrast, structure-preserving architectures exactly impose physics by construction via carefully designed networks. Several works have shown that penalty-based approaches suffer in comparison, with structure-preservation providing improved long term stability, extrapolation and physical realizability.

**Structure-preserving graph networks:** Several works use discretizations of specific PDEs to combat oversmoothing or exploding/vanishing gradients, e.g. telegraph equations [44] or various reaction-diffusion systems [45]. Several works develop Hamiltonian flows on graphs [46, 47]. For metriplectic dynamics, [48] poses a penalty based formulation on graphs. We particularly focus on GRAND, which poses graph learning as a diffusive process [49], using a similar exterior calculus framework and interpreting attention as a diffusion coefficient. We show in Appendix A.5 that their

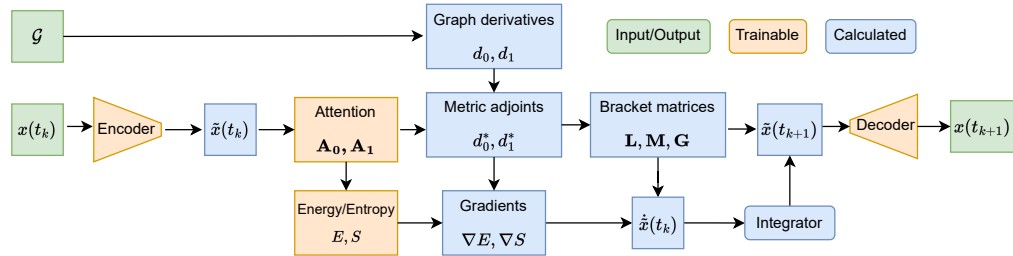

Figure 1: A diagrammatic illustration of the bracket-based architectures introduced in Section 4.

analysis fails to account for the asymmetry in the attention mechanism, leading to a departure from the governing theory. To account for this, we introduce a *modified attention mechanism* which retains interpretation as a part of diffusion PDE. In this purely irreversible case, it is of interest whether adherence to the theory provides improved results, or GRAND's success is driven by something other than structure-preservation.

## 3  Theory and fundamentals

Here we introduce the two essential ingredients to our approach: bracket-based dynamical systems for neural differential equations, and the data-driven exterior calculus which enables their construction. A thorough introduction to this material is provided in Appendices A.1, A.3, and A.2.

**Bracket-based dynamics:** Originally introduced as an extension of Hamiltonian/Lagrangian dynamics to include dissipation [50], bracket formulations are used to inform a dynamical system with certain structural properties, e.g., time-reversibility, invariant differential forms, or property preservation. Even without dissipation, bracket formulations may compactly describe dynamics while preserving core mathematical properties, making them ideal for designing neural architectures.

Bracket formulations are usually specified via some combination of reversible brackets $\{F, G\} = \langle \nabla F, \mathbf{L} \nabla G \rangle$ and irreversible brackets $[F, G] = \langle \nabla F, \mathbf{M} \nabla G \rangle$, $\{\{F, G\}\} = \langle \nabla F, \mathbf{L}^2 \nabla G \rangle$ for potentially state-dependent operators $\mathbf{L}^* = -\mathbf{L}$ and $\mathbf{M}^* = \mathbf{M}$. The particular brackets which are used in the present network architectures are summarized in Table 1. Note that complete systems are the dynamical extensions of isolated thermodynamical systems: they conserve energy and produce entropy, with nothing lost to the ambient environment. Conversely, incomplete systems do not account for any lost energy: they only require that it vanish in a prescribed way. The choice of completeness is an application-dependent modeling assumption.

**Exterior calculus:** In the combinatorial Hodge theory [51], an oriented graph $\mathcal{G} = \{\mathcal{V}, \mathcal{E}\}$ carries sets of $k$-cliques, denoted $\mathcal{G}_k$, which are collections of ordered subgraphs generated by $(k+1)$ nodes. This induces natural exterior derivative operators $d_k : \Omega_k \to \Omega_{k+1}$, acting on the spaces of functions on $\mathcal{G}_k$, which are the signed incidence matrices between $k$-cliques and $(k+1)$-cliques. An explicit representation of these derivatives is given in Appendix A.1, from which it is easy to check the exact sequence property $d_{k+1} \circ d_k = 0$ for any $k$. This yields a discrete de Rham complex on the graph $\mathcal{G}$ (Figure 2). Moreover, given a choice of inner product (say, $\ell^2$) on $\Omega_k$, there is an obvious dual de Rham complex which comes directly from adjointness. In particular, one can define dual derivatives $d_k^* : \Omega_{k+1} \to \Omega_k$ via the equality

$$\langle d_k f, g \rangle_{k+1} = \langle f, d_k^* g \rangle_k ,$$

from which nontrivial results such as the Hodge decomposition, Poincaré inequality, and coercivity/invertibility of the Hodge Laplacian $\Delta_k = d_k^* d_k + d_{k-1} d_{k-1}^*$ follow (see e.g. [11]). Using the derivatives $d_k, d_k^*$, it is possible to build compatible discretizations of PDEs on $\mathcal{G}$ which are guaranteed to preserve exactness properties such as, e.g., $d_1 \circ d_0 = \mathrm{curl} \circ \mathrm{grad} = 0$.

The choice of inner product $\langle \cdot, \cdot \rangle_k$ thus induces a definition of the dual derivatives $d_k^*$. In the graph setting [52], one typically selects the $\ell^2$ inner product, obtaining the adjoints of the signed incidence matrices as $d_k^* = d_k^\mathsf{T}$. By instead working with the modified inner product $(\mathbf{v}, \mathbf{w}) = \mathbf{v}^\mathsf{T} \mathbf{A}_k \mathbf{w}$ for a machine-learnable $\mathbf{A}_k$, we obtain $d_k^* = \mathbf{A}_k^{-1} d_k^\mathsf{T} \mathbf{A}_{k+1}$ (see Appendix A.3). This parameterization

$$\Omega_0 \underset{d_0^\mathsf{T}}{\overset{d_0}{\rightleftarrows}} \Omega_1 \underset{d_1^\mathsf{T}}{\overset{d_1}{\rightleftarrows}} \Omega_2 \underset{d_2^\mathsf{T}}{\overset{d_2}{\rightleftarrows}} \cdots \underset{d_{k-1}^\mathsf{T}}{\overset{d_{k-1}}{\rightleftarrows}} \Omega_k$$

$$\uparrow \mathbf{A}_0 \qquad \uparrow \mathbf{A}_1 \qquad \uparrow \mathbf{A}_2 \qquad\qquad \uparrow \mathbf{A}_k$$

$$\Omega_0 \xleftarrow{d_0^*} \Omega_1 \xleftarrow{d_1^*} \Omega_2 \xleftarrow{d_2^*} \cdots \xleftarrow{d_{k-1}^*} \Omega_k$$

Figure 2: A commutative diagram illustrating the relationship between the graph derivatives $d_k$, their $\ell^2$ adjoints $d_k^\mathsf{T}$, and the learnable adjoints $d_k^*$. These operators form a *de Rham complex* due to the exact sequence property $d_{i+1} \circ d_i = d_i^\mathsf{T} \circ d_{i+1}^\mathsf{T} = d_i^* \circ d_{i+1}^* = 0$. We show that the learnable $\mathbf{A}_k$ may encode attention mechanisms, without impacting the preservation of exact sequence structure.

inherits the exact sequence property from the graph topology encoded in $d_k$ while allowing for incorporation of geometric information from data. This leads directly to the following result, which holds for any (potentially feature-dependent) symmetric positive definite matrix $\mathbf{A}_k$.

**Theorem 3.1.** *The dual derivatives $d_k^* : \Omega_{k+1} \to \Omega_k$ adjoint to $d_k : \Omega_k \to \Omega_{k+1}$ with respect to the learnable inner products $\mathbf{A}_k : \Omega_k \to \Omega_k$ satisfy an exact sequence property.*

*Proof.* $\quad d_{k-1}^* d_k^* = \mathbf{A}_{k-1}^{-1} d_{k-1}^\mathsf{T} \mathbf{A}_k \mathbf{A}_k^{-1} d_k^\mathsf{T} \mathbf{A}_{k+1} = \mathbf{A}_{k-1}^{-1} (d_k d_{k-1})^\mathsf{T} \mathbf{A}_{k+1} = 0.$ □

As will be shown in Section 4, by encoding graph attention into the $\mathbf{A}_k$, we may exploit the exact sequence property to obtain symmetric positive definite diffusion operators, as well as conduct the cancellations necessary to enforce degeneracy conditions necessary for metriplectic dynamics.

For a thorough review of DDEC, we direct readers to Appendix A.1 and [11]. For exterior calculus in topological data analysis see [52], and an overview in the context of PDEs see [53, 54].

## 4 Structure-preserving bracket parameterizations

We next summarize properties of the bracket dynamics introduced in Section 3 and displayed in Table 1, postponing details and rigorous discussion to Appendices A.3 and A.6. Letting $\mathbf{x} = (\mathbf{q}, \mathbf{p})$ denote node-edge feature pairs, the following operators will be used to generate our brackets.

$$\mathbf{L} = \begin{pmatrix} 0 & -d_0^* \\ d_0 & 0 \end{pmatrix}, \quad \mathbf{G} = \begin{pmatrix} \Delta_0 & 0 \\ 0 & \Delta_1 \end{pmatrix} = \begin{pmatrix} d_0^* d_0 & 0 \\ 0 & d_1^* d_1 + d_0 d_0^* \end{pmatrix}, \quad \mathbf{M} = \begin{pmatrix} 0 & 0 \\ 0 & \mathbf{A}_1 d_1^* d_1 \mathbf{A}_1 \end{pmatrix}.$$

As mentioned before, the inner products $\mathbf{A}_0, \mathbf{A}_1, \mathbf{A}_2$ on $\Omega_k$ which induce the dual derivatives $d_0^*, d_1^*$, are chosen in such a way that their combination generalizes a graph attention mechanism. The precise details of this construction are given below, and its relationship to the standard GAT network from [12] is shown in Appendix A.5. Notice that $\mathbf{L}^* = -\mathbf{L}$, while $\mathbf{G}^* = \mathbf{G}, \mathbf{M}^* = \mathbf{M}$ are positive semi-definite with respect to the block-diagonal inner product $(\cdot, \cdot)$ defined by $\mathbf{A} = \mathrm{diag}\,(\mathbf{A}_0, \mathbf{A}_1)$ (details are provided in Appendix A.6). Therefore, $\mathbf{L}$ generates purely reversible (Hamiltonian) dynamics and $\mathbf{G}, \mathbf{M}$ generate irreversible (dissipative) ones. Additionally, note that state-dependence in $\mathbf{L}, \mathbf{M}, \mathbf{G}$ enters only through the adjoint differential operators, meaning that any structural properties induced by the topology of the graph $\mathcal{G}$ (such as the exact sequence property mentioned in Theorem 3.1) are automatically preserved.

**Remark 4.1.** *Strictly speaking, $\mathbf{L}$ is guaranteed to be a truly Hamiltonian system only when $d_0^*$ is state-independent, since it may otherwise fail to satisfy Jacobi's identity. On the other hand, energy conservation is always guaranteed due to the fact that $\mathbf{L}$ is skew-adjoint.*

In addition to the bracket matrices $\mathbf{L}, \mathbf{M}, \mathbf{G}$, it is necessary to have access to energy and entropy functions $E, S$ and their associated functional derivatives with respect to the inner product on $\Omega_0 \oplus \Omega_1$ defined by $\mathbf{A}$. For the Hamiltonian, gradient, and double brackets, $E$ is chosen simply as the "total kinetic energy"

$$E(\mathbf{q}, \mathbf{p}) = \frac{1}{2}\left(|\mathbf{q}|^2 + |\mathbf{p}|^2\right) = \frac{1}{2}\sum_{i \in \mathcal{V}} |\mathbf{q}_i|^2 + \frac{1}{2}\sum_{\alpha \in \mathcal{E}} |\mathbf{p}_\alpha|^2,$$

whose $\mathbf{A}$-gradient (computed in Appendix A.6) is just $\nabla E(\mathbf{q}, \mathbf{p}) = \left( \mathbf{A}_0^{-1}\mathbf{q} \quad \mathbf{A}_1^{-1}\mathbf{p} \right)^\mathsf{T}$. Since the metriplectic bracket uses parameterizations of $E, S$ which are more involved, discussion of this case is deferred to later in this Section.

**Attention as learnable inner product:** Before describing the dynamics, it remains to discuss how the matrices $\mathbf{A}_i$, $0 \leq i \leq 2$, are computed in practice, and how they relate to the idea of graph attention. Recall that if $n_V > 0$ denotes the nodal feature dimension, a graph attention mechanism takes the form $a(\mathbf{q}_i, \mathbf{q}_j) = f\left(\tilde{a}_{ij}\right) / \sum_j f\left(\tilde{a}_{ij}\right)$ for some differentiable pre-attention function $\tilde{a} : n_V \times n_V \to \mathbb{R}$ (e.g., for scaled dot product [55]) one typically represents $a(\mathbf{q}_i, \mathbf{q}_j)$ as a softmax, so that $f = \exp(\mathbf{q})$). This suggests a decomposition $a(\mathbf{q}_i, \mathbf{q}_j) = \mathbf{A}_0^{-1}\mathbf{A}_1$ where $\mathbf{A}_0 = (a_{0,ii})$ is diagonal on nodes and $\mathbf{A}_1 = (a_{1,ij})$ is diagonal on edges,

$$a_{0,ii} = \sum_{j \in \mathcal{N}(i)} f\left(\tilde{a}\left(\mathbf{q}_i, \mathbf{q}_j\right)\right), \qquad a_{1,ij} = f\left(\tilde{a}\left(\mathbf{q}_i, \mathbf{q}_j\right)\right).$$

Treating the numerator and denominator of the standard attention mechanism separately in $\mathbf{A}_0, \mathbf{A}_1$ allows for a flexible and theoretically sound incorporation of graph attention directly into the adjoint differential operators on $\mathcal{G}$. In particular, *if $\mathbf{A}_1$ is symmetric* with respect to edge-orientation and $\mathbf{p}$ is an edge feature which is antisymmetric, it follows that

$$(d_0^*\mathbf{p})_i = \left( \mathbf{A}_0^{-1} d_0^\mathsf{T} \mathbf{A}_1 \mathbf{p} \right)_i = \sum_{j \in \mathcal{N}(i)} a\left(\mathbf{q}_i, \mathbf{q}_j\right) \mathbf{p}_{ji},$$

which is just graph attention combined with edge aggregation. This makes it possible to give the following informal statement regarding graph attention networks which is explained and proven in Appendix A.5.

**Remark 4.2.** *The GAT layer from [12] is almost the forward Euler discretization of a metric heat equation.*

The "almost" appearing here has to do with the fact that (1) the attentional numerator $f\left(\tilde{a}(\mathbf{q}_i, \mathbf{q}_j)\right)$ is generally asymmetric in $i, j$, and is therefore symmetrized by the divergence operator $d_0^\mathsf{T}$, (2) the activation function between layers is not included, and (3) learnable weight matrices $\mathbf{W}^k$ in GAT are set to the identity.

**Remark 4.3.** *The interpretation of graph attention as a combination of learnable inner products admits a direct generalization to higher-order cliques, which is discussed in Appendix A.4.*

**Hamiltonian case:** A purely conservative system is generated by solving $\dot{\mathbf{x}} = \mathbf{L}(\mathbf{x})\nabla E(\mathbf{x})$, or

$$\begin{pmatrix} \dot{\mathbf{q}} \\ \dot{\mathbf{p}} \end{pmatrix} = \begin{pmatrix} 0 & -d_0^* \\ d_0 & 0 \end{pmatrix} \begin{pmatrix} \mathbf{A}_0^{-1} & 0 \\ 0 & \mathbf{A}_1^{-1} \end{pmatrix} \begin{pmatrix} \mathbf{q} \\ \mathbf{p} \end{pmatrix} = \begin{pmatrix} -d_0^* \mathbf{A}_1^{-1} \mathbf{p} \\ d_0 \mathbf{A}_0^{-1} \mathbf{q} \end{pmatrix}.$$

This is a noncanonical Hamiltonian system which generates a purely reversible flow. In particular, it can be shown that

$$\dot{E}(\mathbf{x}) = (\dot{\mathbf{x}}, \nabla E(\mathbf{x})) = (\mathbf{L}(\mathbf{x})\nabla E(\mathbf{x}), \nabla E(\mathbf{x})) = -(\nabla E(\mathbf{x}), \mathbf{L}(\mathbf{x})\nabla E(\mathbf{x})) = 0,$$

so that energy is conserved due to the skew-adjointness of $\mathbf{L}$.

**Gradient case:** On the opposite end of the spectrum are generalized gradient flows, which are totally dissipative. Consider solving $\dot{\mathbf{x}} = -\mathbf{G}(\mathbf{x})\nabla E(\mathbf{x})$, or

$$\begin{pmatrix} \dot{\mathbf{q}} \\ \dot{\mathbf{p}} \end{pmatrix} = - \begin{pmatrix} \Delta_0 & 0 \\ 0 & \Delta_1 \end{pmatrix} \begin{pmatrix} \mathbf{A}_0^{-1} & 0 \\ 0 & \mathbf{A}_1^{-1} \end{pmatrix} \begin{pmatrix} \mathbf{q} \\ \mathbf{p} \end{pmatrix} = - \begin{pmatrix} \Delta_0 \mathbf{A}_0^{-1} \mathbf{q} \\ \Delta_1 \mathbf{A}_1^{-1} \mathbf{p} \end{pmatrix}.$$

This system is a metric diffusion process on nodes and edges separately. Moreover, it corresponds to a generalized gradient flow, since

$$\dot{E}(\mathbf{x}) = (\dot{\mathbf{x}}, \nabla E(\mathbf{x})) = -(\mathbf{G}(\mathbf{x})\nabla E(\mathbf{x}), \nabla E(\mathbf{x})) = -|\nabla E(\mathbf{x})|_{\mathbf{G}}^2 \leq 0,$$

due to the self-adjoint and positive semi-definite nature of $\mathbf{G}$.

**Remark 4.4.** *The architecture in GRAND [49] is almost a gradient flow, however the pre-attention mechanism lacks the requisite symmetry to formally induce a valid inner product.*

**Double bracket case:** Another useful formulation for incomplete systems is the so-called double-bracket formalism. Consider solving $\dot{\mathbf{x}} = \mathbf{L}\nabla E + \mathbf{L}^2\nabla E$, or

$$\begin{pmatrix} \dot{\mathbf{q}} \\ \dot{\mathbf{p}} \end{pmatrix} = \begin{pmatrix} 0 & -d_0^* \\ d_0 & 0 \end{pmatrix} \begin{pmatrix} \mathbf{A}_0^{-1}\mathbf{q} \\ \mathbf{A}_1^{-1}\mathbf{p} \end{pmatrix} + \begin{pmatrix} -d_0^* d_0 & 0 \\ 0 & -d_0 d_0^* \end{pmatrix} \begin{pmatrix} \mathbf{A}_0^{-1}\mathbf{q} \\ \mathbf{A}_1^{-1}\mathbf{p} \end{pmatrix} = \begin{pmatrix} -\Delta_0 \mathbf{A}_0^{-1}\mathbf{q} - d_0^* \mathbf{A}_1^{-1}\mathbf{p} \\ d_0 \mathbf{A}_0^{-1}\mathbf{q} - d_0 d_0^* \mathbf{A}_1^{-1}\mathbf{p} \end{pmatrix}.$$

This provides a dissipative relationship which preserves the Casimirs of the Poisson bracket generated by $\mathbf{L}$, since $\mathbf{L}\nabla C = \mathbf{0}$ implies $\mathbf{L}^2\nabla C = \mathbf{0}$. In particular, it follows that

$$\dot{E}(\mathbf{x}) = (\dot{\mathbf{x}}, \nabla E(\mathbf{x})) = \left(\mathbf{L}(\mathbf{x})\nabla E(\mathbf{x}) + \mathbf{L}^2(\mathbf{x})\nabla E(\mathbf{x}), \nabla E(\mathbf{x})\right) = 0 - |\mathbf{L}(\mathbf{x})\nabla E(\mathbf{x})|^2 \leq 0,$$

since $\mathbf{L}$ is skew-adjoint and therefore $\mathbf{L}^2$ is self-adjoint.

**Remark 4.5.** *It is interesting to note that the matrix $\mathbf{L}$ is essentially a Dirac operator (square root of the Hodge Laplacian $\Delta = (d + d^*)^2$) restricted to cliques of degree at most 1. However, here $\mathbf{L}^2 = -\Delta$, so that $\mathbf{L}$ is in some sense "pure imaginary".*

**Metriplectic case:** Metriplectic systems are expressible as $\dot{\mathbf{x}} = \mathbf{L}\nabla E + \mathbf{M}\nabla S$ where $E, S$ are energy resp. entropy functions which satisfy the degeneracy conditions $\mathbf{L}\nabla S = \mathbf{M}\nabla E = \mathbf{0}$. One way of setting this up in the present case is to define the energy and entropy functions

$$E(\mathbf{q}, \mathbf{p}) = f_E\left(s(\mathbf{q})\right) + g_E\left(s\left(d_0 d_0^\intercal \mathbf{p}\right)\right),$$
$$S(\mathbf{q}, \mathbf{p}) = g_S\left(s\left(d_1^\intercal d_1 \mathbf{p}\right)\right),$$

where $s$ is sum aggregation over nodes resp. edges, $f_E : \mathbb{R}^{n_V} \to \mathbb{R}$ acts on node features, and $g_E, g_S : \mathbb{R}^{n_E} \to \mathbb{R}$ act on edge features. Denoting the "all ones" vector (of variable length) by $\mathbf{1}$, it is shown in Appendix A.6 that the $\mathbf{A}$-gradients of energy and entropy can be computed as

$$\nabla E(\mathbf{x}) = \begin{pmatrix} \mathbf{A}_0^{-1}\mathbf{1} \otimes \nabla f_E\left(h(\mathbf{q})\right) \\ \mathbf{A}_1^{-1} d_0 d_0^\intercal \mathbf{1} \otimes \nabla g_E\left(h\left(d_0 d_0^\intercal \mathbf{p}\right)\right) \end{pmatrix}, \qquad \nabla S(\mathbf{x}) = \begin{pmatrix} 0 \\ \mathbf{A}_1^{-1} d_1^\intercal d_1 \mathbf{1} \otimes \nabla g_S\left(h\left(d_1^\intercal d_1 \mathbf{p}\right)\right) \end{pmatrix}.$$

Similarly, it is shown in Appendix A.6 that the degeneracy conditions $\mathbf{L}\nabla S = \mathbf{M}\nabla E = \mathbf{0}$ are satisfied by construction. Therefore, the governing dynamical system becomes

$$\begin{pmatrix} \dot{\mathbf{q}} \\ \dot{\mathbf{p}} \end{pmatrix} = \mathbf{L}\nabla E + \mathbf{M}\nabla S = \begin{pmatrix} -\mathbf{A}_0^{-1} d_0^\intercal d_0 d_0^\intercal \mathbf{1} \otimes \nabla g_E\left(s\left(d_0 d_0^\intercal \mathbf{p}\right)\right) \\ d_0 \mathbf{A}_0^{-1}\mathbf{1} \otimes \nabla f_E\left(s(\mathbf{q})\right) + \mathbf{A}_1 d_1^* d_1 d_1^\intercal d_1 \mathbf{1} \otimes \nabla g_S\left(s\left(d_1^\intercal d_1 \mathbf{p}\right)\right) \end{pmatrix}.$$

With this, it follows that the system obeys a version of the first and second laws of thermodynamics,

$$\dot{E}(\mathbf{x}) = (\dot{\mathbf{x}}, \nabla E(\mathbf{x})) = (\mathbf{L}\nabla E(\mathbf{x}), \nabla E(\mathbf{x})) + (\mathbf{M}\nabla S(\mathbf{x}), \nabla E(\mathbf{x})) = (\nabla S(\mathbf{x}), \mathbf{M}\nabla E(\mathbf{x})) = 0,$$
$$\dot{S}(\mathbf{x}) = (\dot{\mathbf{x}}, \nabla S(\mathbf{x})) = (\mathbf{L}\nabla E(\mathbf{x}), \nabla S(\mathbf{x})) + (\mathbf{M}\nabla S(\mathbf{x}), \nabla S(\mathbf{x})) = 0 + |\nabla S(\mathbf{x})|_M^2 \geq 0.$$

**Remark 4.6.** *As seen in the increased complexity of this formulation, enforcing the degeneracy conditions necessary for metriplectic structure is nontrivial. This is accomplished presently via an application of the exact sequence property in Theorem 3.1, which we derive in Appendix A.6.*

**Remark 4.7.** *It is worth mentioning that, similar to the other architectures presented in this Section, the metriplectic network proposed here exhibits linear $O(N)$ scaling in the graph size. This is in notable contrast to [29, 31] which scale as $O(N^3)$.*

## 5 Experiments

This section reports results on experiments designed to probe the influence of bracket structure on trajectory prediction and nodal feature classification. Additional experimental details can be found in Appendix B. In each Table, orange indicates the best result by our models, and blue indicates the best of those compared. We consider both physical systems, where the role of structure preservation is explicit, as well as graph-analytic problems.

### 5.1 Damped double pendulum

As a first experiment, consider applying one of these architectures to reproduce the trajectory of a double pendulum with a damping force proportional to the angular momenta of the pendulum masses (see Appendix B.1 for details).

| Double pendulum | MAE q | MAE p | Total MAE |
|---|---|---|---|
| **NODE** | $0.0240 \pm 0.015$ | $0.0299 \pm 0.0091$ | $0.0269 \pm 0.012$ |
| **NODE+AE** | $0.0532 \pm 0.029$ | $0.0671 \pm 0.043$ | $0.0602 \pm 0.035$ |
| **Hamiltonian** | $0.00368 \pm 0.0015$ | $0.00402 \pm 0.0015$ | $0.00369 \pm 0.0013$ |
| **Gradient** | $0.00762 \pm 0.0023$ | $0.0339 \pm 0.012$ | $0.0208 \pm 0.0067$ |
| **Double Bracket** | $0.00584 \pm 0.0013$ | $0.0183 \pm 0.0071$ | $0.0120 \pm 0.0037$ |
| **Metriplectic** | $0.00364 \pm 0.00064$ | $0.00553 \pm 0.00029$ | $0.00459 \pm 0.00020$ |

Table 2: Mean absolute errors (MAEs) of the network predictions in the damped double pendulum case, reported as avg±stdev over 5 runs.

Since this system is metriplectic when expressed in position-momentum-entropy coordinates (c.f. [56]), it is useful to see if any of the brackets from Section 4 can adequately capture these dynamics without an entropic variable. The results of applying the architectures of Section 4 to reproduce a trajectory of five periods are displayed in Table 2, alongside comparisons with a black-box NODE network and a latent NODE with feature encoder/decoder. While each network is capable of producing a small mean absolute error, it is clear that the metriplectic and Hamiltonian networks produce the most accurate trajectories. It is remarkable both that the Hamiltonian bracket does so well here and that the gradient bracket does so poorly, being that the damped double pendulum system is quite dissipative. On the other hand, it is unlikely to be only the feature encoder/decoder leading to good performance here, as both the NODE and NODE+AE architectures perform worse on this task by about one order of magnitude.

## 5.2 MuJoCo Dynamics

Next we test the proposed models on more complex physical systems that are generated by the Multi-Joint dynamics with Contact (MuJoCo) physics simulator [57]. We consider the modified versions of Open AI Gym environments [23]: HalfCheetah, Hopper, and Swimmer.

We represent an object in an environment as a fully-connected graph, where a node corresponds to a body part of the object and, thus, the nodal feature $q_i$ corresponds to a position of a body part or an angle of a joint.[3] As the edge features, a pair of nodal velocities $p_\alpha = (v_{\mathrm{src}(\alpha)}, v_{\mathrm{dst}(\alpha)})$ are provided, where $v_{\mathrm{src}(\alpha)}$ and $v_{\mathrm{dst}(\alpha)}$ denote velocities of the source and destination nodes connected to the edge.

Since the MuJoCo environments contain an actor applying controls, additional control input is accounted for with an additive forcing term which is parameterized by a multi-layer perceptron and introduced into the bracket-based dynamics models. See Appendix B.2 for additional experimental details. The problem therefore consists of finding an optimal control MLP, and we evaluate the improvement which comes from representing the physics surrogate with bracket dynamics over NODE.

All models are trained via minimizing the MSE between the predicted positions $\tilde{q}$ and the ground truth positions $q$ and are tested on an unseen test set. Table 3 reports the errors of network predictions on the test set measured in the relative $\ell_2$ norm, $\|q - \tilde{q}\|_2 / \|q\|_2 \|\tilde{q}\|_2$. Similar to the double pendulum experiments, all models are able to produce accurate predictions with around or less than $10\%$ errors. While the gradient bracket makes little to no improvements over NODEs, the Hamiltonian, double, and metriplectic brackets produce more accurate predictions. Interestingly, the Hamiltonian bracket performs the best in this case as well, meaning that any dissipation present is effectively compensated for by the autoencoder which transforms the features.

## 5.3 Node classification

Moving beyond physics-based examples, it remains to see how bracket-based architectures perform on "black-box" node classification problems. Table 4 and Table 5 present results on common benchmark problems including the citation networks Cora [58], Citeseer [59], and Pubmed [60], as well as the coauthor graph, CoauthorCS [61], and the Amazon co-purchasing graphs, Computer and Photo [62]. For comparison, we report results on a standard GAT [12], a neural graph differential equation

---

[3]Results of an experiment with an alternative embedding (i.e., $q_i = (q_i, v_i)$) are reported in Appendix B.2.2.

| Dataset | HalfCheetah | Hopper | Swimmer |
|---|---|---|---|
| **NODE+AE** | $0.106 \pm 0.0011$ | $0.0780 \pm 0.0021$ | $0.0297 \pm 0.0036$ |
| **Hamiltonian** | $0.0566 \pm 0.013$ | $0.0279 \pm 0.0019$ | $0.0122 \pm 0.00044$ |
| **Gradient** | $0.105 \pm 0.0076$ | $0.0848 \pm 0.0011$ | $0.0290 \pm 0.0011$ |
| **Double Bracket** | $0.0621 \pm 0.0096$ | $0.0297 \pm 0.0048$ | $0.0128 \pm 0.00070$ |
| **Metriplectic** | $0.105 \pm 0.0091$ | $0.0398 \pm 0.0057$ | $0.0179 \pm 0.00059$ |

Table 3: Relative error of network predictions for the MuJoCo environment on the test set, reported as avg±stdev over 4 runs.

| Planetoid splits | CORA | CiteSeer | PubMed |
|---|---|---|---|
| **GAT** | $82.8 \pm 0.5$ | $69.5 \pm 0.9$ | $79.0 \pm 0.5$ |
| **GDE** | $83.8 \pm 0.5$ | $72.5 \pm 0.5$ | $79.9 \pm 0.3$ |
| **GRAND-nl** | $83.6 \pm 0.5$ | $70.8 \pm 1.1$ | $79.7 \pm 0.3$ |
| **Hamiltonian** | $77.2 \pm 0.7$ | $73.0 \pm 1.2$ | $78.5 \pm 0.3$ |
| **Gradient** | $79.9 \pm 0.7$ | $71.8 \pm 1.4$ | $78.6 \pm 0.7$ |
| **Double Bracket** | $82.6 \pm 0.9$ | $74.2 \pm 1.4$ | $79.6 \pm 0.6$ |
| **Metriplectic** | $57.4 \pm 1.0$ | $60.5 \pm 1.1$ | $69.8 \pm 0.7$ |

Table 4: Test accuracy and standard deviations (averaged over 20 randomly initialized runs) using the original Planetoid train-valid-test splits. Comparisons use the numbers reported in [49].

architecture (GDE) [63], and the nonlinear GRAND architecture (GRAND-nl) from [49] which is closest to ours. Since our experimental setting is similar to that of [49], the numbers reported for GAT, GDE, and GRAND-nl are taken directly from this paper. Note that, despite the similar $O(N)$ scaling in the metriplectic architecture, the high dimension of the node and edge features on the latter three datasets led to trainable $E, S$ functions which exhausted the memory on our available machines, and therefore results are not reported for these cases. A full description of experimental details is provided in Appendix B.3.

**Remark 5.1.** *To highlight the effect of bracket structure on network performance, only minimal modifications are employed during network training. In particular, we do not include any additional regularization, positional encoding, graph rewiring, extraction of connected components, extra terms on the right-hand side, or early stopping. While it is likely that better classification performance could be achieved with some of these modifications included, it becomes very difficult to isolate the effect of structure-preservation. A complete list of tunable hyperparameters is given in Appendix B.3.*

The results show different behavior produced by each bracket architecture. It is empirically clear that there is some value in full or partial reversibility, since the Hamiltonian and double bracket architectures both perform better than the corresponding gradient architecture on datasets such as Computer and Photo. Moreover, it appears that the partially reversible double bracket performs the best of the bracket architectures in every case, which is consistent with the idea that both reversible and irreversible dynamics are critical for capturing the behavior of general dynamical systems. Interestingly, the metriplectic bracket performs worse on these tasks by a large margin. We conjecture this architecture may be harder to train for larger problems despite its $O(N)$ complexity in the graph size, suggesting that more sophisticated training strategies may be required for large problems.

# 6 Conclusion

This work presents a unified theoretical framework for analysis and construction of graph attention networks. The exact sequence property of graph derivatives and coercivity of Hodge Laplacians which follow from the theory allow the construction of four structure-preserving brackets, which we use to evaluate the role of irreversibility in both data-driven physics simulators and graph analytics problems. In all contexts, the pure diffusion bracket performed most poorly, with mixed results between purely reversible and partially dissipative brackets.

| Random splits | CORA | CiteSeer | PubMed | Coauthor CS | Computer | Photo |
|---|---|---|---|---|---|---|
| **GAT** | $81.8 \pm 1.3$ | $71.4 \pm 1.9$ | $78.7 \pm 2.3$ | $90.5 \pm 0.6$ | $78.0 \pm 19.0$ | $85.7 \pm 20.3$ |
| **GDE** | $78.7 \pm 2.2$ | $71.8 \pm 1.1$ | $73.9 \pm 3.7$ | $91.6 \pm 0.1$ | $82.9 \pm 0.6$ | $92.4 \pm 2.0$ |
| **GRAND-nl** | $82.3 \pm 1.6$ | $70.9 \pm 1.0$ | $77.5 \pm 1.8$ | $92.4 \pm 0.3$ | $82.4 \pm 2.1$ | $92.4 \pm 0.8$ |
| **Hamiltonian** | $76.2 \pm 2.1$ | $72.2 \pm 1.9$ | $76.8 \pm 1.1$ | $92.0 \pm 0.2$ | $84.0 \pm 1.0$ | $91.8 \pm 0.2$ |
| **Gradient** | $81.3 \pm 1.2$ | $72.1 \pm 1.7$ | $77.2 \pm 2.1$ | $92.2 \pm 0.3$ | $78.1 \pm 1.2$ | $88.2 \pm 0.6$ |
| **Double Bracket** | $83.0 \pm 1.1$ | $74.2 \pm 2.5$ | $78.2 \pm 2.0$ | $92.5 \pm 0.2$ | $84.8 \pm 0.5$ | $92.4 \pm 0.3$ |
| **Metriplectic** | $59.6 \pm 2.0$ | $63.1 \pm 2.4$ | $69.8 \pm 2.1$ | - | - | - |

Table 5: Test accuracy and standard deviations averaged over 20 runs with random 80/10/10 train/val/test splits. Comparisons use the numbers reported in [49].

The linear scaling achieved by the metriplectic brackets has a potential major impact for data-driven physics modeling. Metriplectic systems emerge naturally when coarse-graining multiscale systems. With increasing interest in using ML to construct digital twins, fast data-driven surrogates for complex multi-physics acting over multiple scales will become crucial. In this setting the stability encoded by metriplectic dynamics translates to robust surrogates, with linear complexity suggesting the possibility of scaling up to millions of degrees of freedom.

**Limitations:** All analysis holds under the assumption of modified attention mechanisms which allow interpretation of GAT networks as diffusion processes; readers should take care that the analysis is for a non-standard attention. Secondly, for all brackets we did not introduce empirical modifications (e.g. regularization, forcing, etc) to optimize performance so that we could study the role of (ir)reversibility in isolation. With this in mind, one may be able to add "tricks" to e.g. obtain a diffusion architecture which outperforms those presented here. Finally, note that the use of a feature autoencoder in the bracket architectures means that structure is enforced in the transformed space. This allows for applicability to more general systems, and can be easily removed when appropriate features are known.

**Broader impacts:** The work performed here is strictly foundational mathematics and is intended to improve the performance of GNNs in the context of graph analysis and data-driven physics modeling. Subsequent application of the theory may have societal impact, but the current work anticipated to improve the performance of machine learning in graph settings only at a foundational level.

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

## Glossary of Notation and Symbols

The next list describes several symbols that will be later used within the body of the document

$\{\{\cdot,\cdot\}\}$  (Irreversible) double bracket on functions with generator $\mathbf{L}^2$

$[\cdot,\cdot]$  Degenerate (irreversible) metric bracket on functions with generator $\mathbf{M}^\mathsf{T} = \mathbf{M}$

$\{\cdot,\cdot\}$  Poisson (reversible) bracket on functions with generator $\mathbf{L}^\mathsf{T} = -\mathbf{L}$

$\mathcal{N}(i), \overline{\mathcal{N}}(i)$  Neighbors of node $i \in \mathcal{V}$, neighbors of node $i \in \mathcal{V}$ including $i$

$[S]$  Indicator function of the statement $S$

$\delta f, \nabla f$  Adjoint of $df$ with respect to $\langle\cdot,\cdot\rangle$, adjoint of $df$ with respect to $(\cdot,\cdot)$

$\Delta_k$  Hodge Laplacian $d_k d_k^* + d_k^* d_k$

$\delta_{ij}$  Kronecker delta

$\dot{f}$  Derivative of $f$ with respect to time

$(\cdot,\cdot)_k$  Learnable metric inner product on $k$-cliques with matrix representation $\mathbf{A}_k$

$\langle\cdot,\cdot\rangle_k$  Euclidean $\ell^2$ inner product on $k$-cliques

$\mathcal{G}, \mathcal{V}, \mathcal{E}$  Oriented graph, set of nodes, set of edges

$\mathcal{G}_k, \Omega_k$  Set of $k$-cliques, vector space of real-valued functions on $k$-cliques

$d, d_k$  Exterior derivative operator on functions, exterior derivative operator on $k$-cliques

$d_k^\mathsf{T}, d_k^*$  Adjoint of $d_k$ with respect to $\langle\cdot,\cdot\rangle_k$, adjoint of $d_k$ with respect to $(\cdot,\cdot)_k$

## A  Mathematical foundations

This Appendix provides the following: (1) an introduction to the ideas of graph exterior calculus, A.1, and bracket-based dynamical systems, A.2, necessary for understanding the results in the body, (2) additional explanation regarding adjoints with respect to generic inner products and associated computations, A.3, (3) a mechanism for higher-order attention expressed in terms of learnable inner products, A.4, (4) a discussion of GATs in the context of exterior calculus, A.5, and (5) proofs which are deferred from Section 4, A.6.

### A.1  Graph exterior calculus

Here some basic notions from the graph exterior calculus are recalled. More details can be found in, e.g., [11, 51, 64].

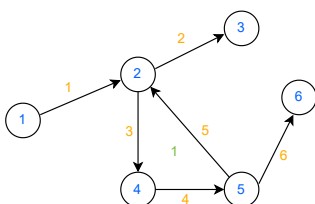

Figure 3: A toy graph with six 0-cliques (nodes), six 1-cliques (edges), and one 2-clique.

As mentioned in Section 3, an oriented graph $\mathcal{G} = \{\mathcal{V}, \mathcal{E}\}$ carries sets of $k$-*cliques*, denoted $\mathcal{G}_k$, which are collections of ordered subgraphs generated by $(k+1)$ nodes. For example, the graph in Figure 3 contains six 0-cliques (nodes), six 1-cliques (edges), and one 2-clique. A notion of combinatorial derivative is then given by the *signed incidence matrices* $d_k : \Omega_k \to \Omega_{k+1}$, operating on the space $\Omega_k$ of differentiable functions on $k$-cliques, whose entries $(d_k)_{ij}$ are 1 or -1 if the $j^{\text{th}}$ $k$-clique is

incident on the $i^{\text{th}}$ $(k+1)$-clique, and zero otherwise. For the example in Figure 3, these are:

$$d_0 = \begin{pmatrix} -1 & 1 & 0 & 0 & 0 & 0 \\ 0 & -1 & 1 & 0 & 0 & 0 \\ 0 & -1 & 0 & 1 & 0 & 0 \\ 0 & 0 & 0 & -1 & 1 & 0 \\ 0 & 1 & 0 & 0 & -1 & 0 \\ 0 & 0 & 0 & 0 & -1 & 1 \end{pmatrix}, \qquad d_1 = \begin{pmatrix} 0 & 0 & 1 & 1 & 1 & 0 \end{pmatrix}.$$

**Remark A.1.** *While the one-hop neighborhood of node $i$ in $\mathcal{G}$, denoted $\mathcal{N}(i)$, does not include node $i$ itself, many machine learning algorithms employ the extended neighborhood $\overline{\mathcal{N}}(i) = \mathcal{N}(i) \cup \{i\}$. Since this is equivalent to considering the one-hop neighborhood of node $i$ in the self-looped graph $\overline{\mathcal{G}}$, this modification does not change the analysis of functions on graphs.*

It can be shown that the action of these matrices can be conveniently expressed in terms of totally antisymmetric functions $f \in \Omega_k$, via the expression

$$\left(d_k f\right)\left(i_0, i_1, ..., i_{k+1}\right) = \sum_{j=0}^{k+1} (-1)^j f\left(i_0, ..., \widehat{i}_j, ..., i_{k+1}\right),$$

where $(i_0, ..., i_{k+1})$ denotes a $(k+1)$-clique of vertices $v \in \mathcal{V}$. As convenient shorthand, we often write subscripts, e.g., $(d_k f)_{i_0 i_1 ... i_{k+1}}$, instead of explicit function arguments. Using $[S]$ to denote the indicator function of the statement $S$, it is straightforward to check that $d \circ d = 0$,

$$\begin{aligned} \left(d_k d_{k-1} f\right)_{i_0, ..., i_{k+1}} &= \sum_{j=0}^{k+1} (-1)^j \left(d_{k-1} f\right)_{i_0, ..., \widehat{i}_j, ..., i_{k+1}} \\ &= \sum_{j=0}^{k+1}\sum_{l=0}^{k+1} [l < j] (-1)^{j+l} f_{i_0...\widehat{i}_l...\widehat{i}_j...i_{k+1}} \\ &\quad + \sum_{j=0}^{k+1}\sum_{l=0}^{k+1} [l > j] (-1)^{j+l-1} f_{i_0...\widehat{i}_j...\widehat{i}_l...i_{k+1}} \\ &= \sum_{l<j} (-1)^{j+l} f_{i_0...\widehat{i}_l...\widehat{i}_j...i_{k+1}} \\ &\quad - \sum_{l<j} (-1)^{j+l} f_{i_0...\widehat{i}_l...\widehat{i}_j...i_{k+1}} = 0, \end{aligned}$$

since $(-1)^{j+l-1} = (-1)^{-1}(-1)^{j+l} = (-1)(-1)^{j+l}$ and the final sum follows from swapping the labels $j, l$. This shows that the $k$-cliques on $\mathcal{G}$ form a *de Rham complex* [53]: a collection of function spaces $\Omega_k$ equipped with mappings $d_k$ satisfying $\text{Im}\, d_{k-1} \subset \text{Ker}\, d_k$ as shown in Figure 4. When

$$\Omega_0 \xrightarrow{\ d_0\ } \Omega_1 \xrightarrow{\ d_1\ } \Omega_2 \xrightarrow{\ d_2\ } \cdots \xrightarrow{\ d_{K-1}\ } \Omega_K$$

Figure 4: Illustration of the de Rham complex on $\mathcal{G}$ induced by the combinatorial derivatives, where $K > 0$ is the maximal clique degree.

$K = 3$, this is precisely the graph calculus analogue of the de Rham complex on $\mathbb{R}^3$ formed by the Sobolev spaces $H^1, H(\text{curl}), H(\text{div}), L^2$ which satisfies $\text{div} \circ \text{curl} = \text{curl} \circ \text{grad} = 0$.

While the construction of the graph derivatives and their associated de Rham complex is purely topological, building elliptic differential operators such as the Laplacian relies on a dual de Rham complex, which is specified by an inner product on $\Omega_k$. In the case of $\ell^2$, this leads to dual derivatives which are the matrix transposes of the $d_k$ having the following explicit expression.

**Proposition A.1.** *The dual derivatives $d_k^\intercal : \Omega_{k+1} \to \Omega_k$ adjoint to $d_k$ through the $\ell^2$ inner product are given by*

$$\left(d_k^\intercal f\right)\left(i_0, i_1, ..., i_k\right) = \frac{1}{k+2} \sum_{i_{k+1}} \sum_{j=0}^{k+1} f\left(i_0, ..., [i_j, ..., i_{k+1}]\right),$$

*where $[i_j, ..., i_{k+1}] = i_{k+1}, i_j, ..., i_k$ indicates a cyclic permutation forward by one index.*

*Proof.* This is a direct calculation using the representation of $d_k$ in terms of antisymmetric functions. More precisely, let an empty sum $\Sigma$ denote summation over all unspecified indices. Then, for any $g \in \Omega_k$,

$$
\begin{aligned}
\langle d_k f, g \rangle &= \sum_{i_0 \dots i_{k+1} \in \mathcal{G}_{k+1}} (d_k f)_{i_0 \dots i_{k+1}} \, g_{i_0 \dots, i_{k+1}} \\
&= \frac{1}{(k+2)!} \sum \left( \sum_{j=0}^{k+1} (-1)^j f_{i_0 \dots \widehat{i_j} \dots i_{k+1}} \right) g_{i_0 \dots i_{k+1}} \\
&= \frac{1}{(k+2)!} \sum f_{i_0 \dots i_k} \left( \sum_{i_{k+1}} \sum_{j=0}^{k+1} (-1)^j g_{i_0 \dots [i_j \dots i_{k+1}]} \right) \\
&= \frac{1}{k+2} \sum_{i_0 i_1 \dots i_k \in \mathcal{G}_k} f_{i_0 \dots i_k} \left( \sum_{i_{k+1}} \sum_{j=0}^{k+1} (-1)^j g_{i_0 \dots [i_j \dots i_{k+1}]} \right) \\
&= \sum_{i_0 i_1 \dots i_k \in \mathcal{G}_k} f_{i_0 \dots i_k} \, (d_k^\mathsf{T} g)_{i_0 \dots i_k} = \langle f, d_k^\mathsf{T} g \rangle,
\end{aligned}
$$

which establishes the result. $\qquad\square$

Proposition A.1 is perhaps best illustrated with a concrete example. Consider the graph gradient, defined for edge $\alpha = (i, j)$ as $(d_0 f)_\alpha = (d_0 f)_{ij} = f_j - f_i$. Notice that this object is antisymmetric with respect to edge orientation, and measures the outflow of information from source to target nodes. From this, it is easy to compute the $\ell^2$-adjoint of $d_0$, known as the graph divergence, via

$$
\begin{aligned}
\langle d_0 f, g \rangle &= \sum_{\alpha=(i,j)} (f_j - f_i) \, g_{ij} = \sum_i \sum_{(j>i) \in \mathcal{N}(i)} g_{ij} f_j - g_{ij} f_i \\
&= \frac{1}{2} \sum_i \sum_{j \in \mathcal{N}(i)} f_i \, (g_{ji} - g_{ij}) = \langle f, d_0^\mathsf{T} g \rangle,
\end{aligned}
$$

where we have re-indexed under the double sum, used that $i \in \mathcal{N}(j)$ if and only if $j \in \mathcal{N}(i)$, and used that there are no self-edges in $\mathcal{E}$. Therefore, it follows that the graph divergence at node $i$ is given by

$$
(d_0^\mathsf{T} g)_i = \sum_{\alpha \ni i} g_{-\alpha} - g_\alpha = \frac{1}{2} \sum_{j \in \mathcal{N}(i)} g_{ji} - g_{ij},
$$

which reduces to the common form $(d_0^\mathsf{T} g)_i = -\sum_j g_{ij}$ if and only if the edge feature $g_{ij}$ is antisymmetric.

**Remark A.2.** *When the inner product on edges $\mathcal{E}$ is not $L^2$, but defined in terms of a nonnegative, orientation-invariant, and (edge-wise) diagonal weight matrix $\mathbf{W} = (w_{ij})$, a similar computation shows that the divergence becomes*

$$
(d_0^* f)_i = \frac{1}{2} \sum_{j \in \mathcal{N}(i)} w_{ij} \, (f_{ji} - f_{ij}).
$$

*The more general case of arbitrary inner products on $\mathcal{V}, \mathcal{E}$ is discussed in section A.3.*

The differential operators $d_k^\mathsf{T}$ induce a dual de Rham complex since $d_{k-1}^\mathsf{T} d_k^\mathsf{T} = (d_k d_{k-1})^\mathsf{T} = 0$, which enables both the construction of Laplace operators on $k$-cliques, $\Delta_k = d_k^\mathsf{T} d_k + d_{k-1} d_{k-1}^\mathsf{T}$, as well as the celebrated Hodge decomposition theorem, stated below. For a proof, see, e.g., [11, Theorem 3.3].

**Theorem A.3.** *(Hodge Decomposition Theorem) The de Rham complexes formed by $d_k, d_k^\mathsf{T}$ induce the following direct sum decomposition of the function space $\Omega_k$,*

$$
\Omega_k = \operatorname{Im} d_{k-1} \oplus \operatorname{Ker} \Delta_k \oplus \operatorname{Im} d_k^\mathsf{T}.
$$

In the case where the dual derivatives $d_k^*$ are adjoint with respect to a learnable inner product which does not depend on graph features, the conclusion of Theorem A.3 continues to hold, leading to an interesting well-posedness result proved in [11] involving nonlinear perturbations of a Hodge-Laplace problem in mixed form.

**Theorem A.4.** *([11, Theorem 3.6]) Suppose $\mathbf{f}_k \in \Omega_k$, and $g(\mathbf{x}; \xi)$ is a neural network with parameters $\xi$ which is Lipschitz continuous and satisfies $g(\mathbf{0}) = \mathbf{0}$. Then, the problem*

$$\mathbf{w}_{k-1} = d_{k-1}^* \mathbf{u}_k + \epsilon g\left(d_{k-1}^* \mathbf{u}_k; \xi\right),$$
$$\mathbf{f}_k = d_{k-1}\mathbf{w}_{k-1} + d_k^* d_k \mathbf{u}_k,$$

*has a unique solution on $\Omega_k / \operatorname{Ker} \Delta_k$.*

This result shows that initial-value problems involving the Hodge-Laplacian are stable under nonlinear perturbations. Moreover, when $\Delta_0$ is the Hodge Laplacian on nodes, there is a useful connection between $\Delta_0$ and the degree and adjacency matrices of the graph $\mathcal{G}$. Recall that the degree matrix $\mathbf{D} = (d_{ij})$ is diagonal with entries $d_{ii} = \sum_{j \in \mathcal{N}(i)} 1$, while the adjacency matrix $\mathbf{A} = (a_{ij})$ satisfies $a_{ij} = 1$ when $j \in \mathcal{N}(i)$ and $a_{ij} = 0$ otherwise.

**Proposition A.2.** *The combinatorial Laplacian on $\mathcal{V}$, denoted $\Delta_0 = d_0^\mathsf{T} d_0$, satisfies $\Delta_0 = \mathbf{D} - \mathbf{A}$.*

*Proof.* Notice that

$$\left(d_0^\mathsf{T} d_0\right)_{ij} = \sum_{\alpha \in \mathcal{E}} (d_0)_{\alpha i} (d_0)_{\alpha j} = [i = j] \sum_{\alpha \in \mathcal{E}} \left((d_0)_{\alpha i}\right)^2 + [i \neq j] \sum_{\alpha = (i,j)} (d_0)_{\alpha i} (d_0)_{\alpha j}$$
$$= [i = j] \, d_{ii} - [i \neq j] \, a_{ij} = d_{ij} - a_{ij} = \mathbf{D} - \mathbf{A},$$

where we used that $\mathbf{D}$ is diagonal, $\mathbf{A}$ is diagonal-free, and $(d_0)_{\alpha i} (d_0)_{\alpha j} = -1$ whenever $\alpha = (i,j)$ is an edge in $\mathcal{E}$, since one of $(d_0)_{\alpha i}, (d_0)_{\alpha j}$ is 1 and the other is -1. $\qquad\square$

## A.2 Bracket-based dynamical systems

Here we mention some additional facts regarding bracket-based dynamical systems. More information can be found in, e.g., [50, 65, 66, 67].

As mentioned before, the goal of bracket formalisms is to extend the Hamiltonian formalism to systems with dissipation. To understand where this originates, consider an action functional $\mathcal{A}(q) = \int_a^b L(q, \dot{q}) \, dt$ on the space of curves $q(t)$, defined in terms of a Lagrangian $L$ on the tangent bundle to some Riemannian manifold. Using $L_q, L_{\dot{q}}$ to denote partial derivatives with respect to the subscripted variable, it is straightforward to show that, for any compactly supported variation $\delta q$ of $q$, we have

$$d\mathcal{A}(q)\delta q = \int_a^b dL(q, \dot{q}) \, \delta q = \int_a^b L_q \delta q + L_{\dot{q}} \delta \dot{q} = \int_a^b \left(L_q - \partial_t L_{\dot{q}}\right) \delta q,$$

where the final equality follows from integration-by-parts and the fact that variational and temporal derivatives commute in this setting. It follows that $\mathcal{A}$ is stationary (i.e., $d\mathcal{A} = 0$) for all variations only when $\partial_t L_{\dot{q}} = L_q$. These are the classical Euler-Lagrange equations which are (under some regularity conditions) transformed to Hamiltonian form via a Legendre transformation,

$$H(q, p) = \sup_{\dot{q}} \left(\langle p, \dot{q}\rangle - L(q, \dot{q})\right),$$

which defines the Hamiltonian functional $\mathcal{H}$ on phase space, and yields the conjugate momentum vector $p = L_{\dot{q}}$. Substituting $L = \langle p, \dot{q}\rangle - H$ into the previously derived Euler-Lagrange equations leads immediately to Hamilton's equations for the state $\mathbf{x} = (q \quad p)^\mathsf{T}$,

$$\dot{\mathbf{x}} = \begin{pmatrix} \dot{q} \\ \dot{p} \end{pmatrix} = \begin{pmatrix} 0 & 1 \\ -1 & 0 \end{pmatrix} \begin{pmatrix} H_q \\ H_p \end{pmatrix} = \mathbf{J}\nabla\mathcal{H},$$

which are an equivalent description of the system in question in terms of the anti-involution $\mathbf{J}$ and the functional gradient $\nabla\mathcal{H}$.

An advantage of the Hamiltonian description is its compact bracket-based formulation, $\dot{\mathbf{x}} = \mathbf{J}\nabla\mathcal{H} = \{\mathbf{x}, \mathcal{H}\}$, which requires only the specification of an antisymmetric Poisson bracket $\{\cdot, \cdot\}$ and a Hamiltonian functional $\mathcal{H}$. Besides admitting a direct generalization to more complex systems such as Korteweg-de Vries or incompressible Euler, where the involved bracket is state-dependent, this formulation makes the energy conservation property of the system obvious. In particular, it follows immediately from the antisymmetry of $\{\cdot, \cdot\}$ that

$$\dot{\mathcal{H}} = \langle \dot{\mathbf{x}}, \nabla\mathcal{H}\rangle = \{\mathcal{H}, \mathcal{H}\} = 0,$$

while it is more difficult to see immediately that the Euler-Lagrange system obeys this same property. The utility and ease-of-use of bracket formulations is what inspired their extension to other systems of interest which do not conserve energy. On the opposite end of this spectrum are the generalized gradient flows, which can be written in terms of a bracket which is purely dissipative. An example of this is heat flow $\dot{q} = \Delta q := -[q, \mathcal{D}]$, which is the $L^2$-gradient flow of Dirichlet energy $\mathcal{D}(q) = (1/2)\int_a^b |q'|^2\, dt$ (c.f. Appendix A.3). In this case, the functional gradient $\nabla\mathcal{D} = -\partial_{tt}$ is the negative of the usual Laplace operator, so that the positive-definite bracket $[\cdot,\cdot]$ is generated by the identity operator $M = \mathrm{id}$. It is interesting to note that the same system could be expressed using the usual kinetic energy $\mathcal{E}(q) = (1/2)\int_a^b |q|^2\, dt$ instead, provided that the corresponding bracket is generated by $M = -\Delta$. This is a good illustration of the flexibility afforded by bracket-based dynamical systems.

Since physical systems are not always purely reversible or irreversible, other useful bracket formalisms have been introduced to capture dynamics which are a mix of these two. The double bracket $\dot{\mathbf{x}} = \{\mathbf{x}, E\} + \{\{\mathbf{x}, E\}\} = \mathbf{L}\nabla E + \mathbf{L}^2\nabla E$ is a nice extension of the Hamiltonian bracket particularly because it is Casimir preserving, i.e., those quantities which annihilate the Poisson bracket $\{\cdot,\cdot\}$ also annihilate the double bracket. This allows for the incorporation of dissipative phenomena into idealized Hamiltonian systems without affecting desirable properties such as mass conservation, and has been used to model, e.g., the Landau-Lifschitz dissipative mechanism, as well as a mechanism for fluids where energy decays but entropy is preserved (see [65] for additional discussion). A complementary but alternative point of view is taken by the metriplectic bracket formalism, which requires that any dissipation generated by the system is accounted for within the system itself through the generation of entropy. In the metriplectic formalism, the equations of motion are $\dot{\mathbf{x}} = \{\mathbf{x}, E\} + [\mathbf{x}, S] = \mathbf{L}\nabla E + \mathbf{M}\nabla S$, along with important and nontrivial compatibility conditions $\mathbf{L}\nabla S = \mathbf{M}\nabla E = \mathbf{0}$, also called degeneracy conditions, which ensure that the reversible and irreversible mechanisms do not cross-contaminate. As shown in the body of the paper, this guarantees that metriplectic systems obey a form of the first and second thermodynamical laws. Practically, the degeneracy conditions enforce a good deal of structure on the operators $\mathbf{L}, \mathbf{M}$ which has been exploited to generate surrogate models [29, 68, 31]. In particular, it can be shown that the reversible and irreversible brackets can be parameterized in terms of a totally antisymmetric order-3 tensor $\boldsymbol{\xi} = (\xi_{ijk})$ and a partially symmetric order-4 tensor $\boldsymbol{\zeta} = (\zeta_{ik,jl})$ through the relations (Einstein summation assumed)

$$\{A, B\} = \xi^{ijk}\,\partial_i A\,\partial_j B\,\partial_k S,$$
$$[A, B] = \zeta^{ik,jl}\,\partial_i A\,\partial_k E\,\partial_j B\,\partial_l E.$$

Moreover, using the symmetries of $\boldsymbol{\zeta}$, it follows (see [67]) that this tensor decomposes into the product $\zeta_{ik,jl} = \Lambda_{ik}^m D_{mn}\Lambda_{jl}^n$ of a symmetric matrix $\mathbf{D}$ and an order-3 tensor $\Lambda$ which is skew-symmetric in its lower indices. Thus, by applying symmetry relationships, it is easy to check that $\{\cdot, S\} = [\cdot, E] = \mathbf{0}$.

**Remark A.5.** *In [29], trainable 4- and 3- tensors $\xi^{ijk}$ and $\zeta^{ik,jl}$ are constructed to achieve the degeneracy conditions, mandating a costly $O(N^3)$ computational complexity. In the current work we overcome this by instead achieving degeneracy through the exact sequence property.*

### A.3  Adjoints and gradients

Beyond the basic calculus operations discussed in section A.1 which depend only on graph topology, the network architectures discussed in the body also make extensive use of learnable metric information coming from the nodal features. To understand this, it is useful to recall some information about general inner products and the derivative operators that they induce. First, recall that the usual $\ell^2$ inner product on node features $\mathbf{a}, \mathbf{b} \in \mathbb{R}^{|\mathcal{V}|}$, $\langle \mathbf{a}, \mathbf{b} \rangle = \mathbf{a}^\mathsf{T}\mathbf{b}$, is (in this context) a discretization of the standard $L^2$ inner product $\int_\mathcal{V} ab\, d\mu$ which aggregates information from across the vertex set $\mathcal{V}$. While this construction is clearly dependent only on the graph structure (i.e., topology), *any* symmetric positive definite (SPD) matrix $\mathbf{A}_0 : \Omega_0 \to \Omega_0$ also defines an inner product on functions $\mathbf{a} \in \Omega_0$ through the equality

$$(\mathbf{a}, \mathbf{b})_0 := \langle \mathbf{a}, \mathbf{A}_0\mathbf{b} \rangle = \mathbf{a}^\mathsf{T}\mathbf{A}_0\mathbf{b},$$

which gives a different way of measuring the distance between $\mathbf{a}$ and $\mathbf{b}$. The advantage of this construction is that $\mathbf{A}_0$ can be chosen in a way that incorporates geometric information which implicitly regularizes systems obeying a variational principle. This follows from the following

intuitive fact: the Taylor series of a function does not change, regardless of the inner product on its domain. For any differentiable function(al) $E : \Omega_0 \to \mathbb{R}$, using $d$ to denote the exterior derivative, this means that the following equality holds

$$dE(\mathbf{a})\mathbf{b} := \lim_{\varepsilon \to 0} \frac{E(\mathbf{a} + \epsilon\mathbf{b}) - E(\mathbf{a})}{\varepsilon} = \langle \delta E(\mathbf{a}), \mathbf{b} \rangle = (\nabla E(\mathbf{a}), \mathbf{b})_0 ,$$

where $\delta E$ denotes the $\ell^2$-gradient of $E$ and $\nabla E$ denotes its $\mathbf{A}_0$-gradient, i.e., its gradient with respect to the derivative operator induced by the inner product involving $\mathbf{A}_0$. From this, it is clear that $\delta E = \mathbf{A}_0 \nabla E$, so that the $\mathbf{A}_0$-gradient is just an anisotropic rescaling of the $\ell^2$ version. The advantage of working with $\nabla$ over $\delta$ in the present case of graph networks is that $\mathbf{A}_0$ can be *learned* based on the features of the graph. This means that learnable feature information (i.e., graph attention) can be directly incorporated into the differential operators governing our bracket-based dynamical systems by construction.

The prototypical example of where this technique is useful is seen in the gradient flow of Dirichlet energy. Recall that the Dirichlet energy of a differentiable function $u : \mathbb{R}^n \to \mathbb{R}$ is given by $\mathcal{D}(u) = (1/2) \int |\nabla u|^2 \, d\mu$, where $\nabla$ now denotes the usual $\ell^2$-gradient of the function $u$ on $\mathbb{R}^n$. Using integration-by-parts, it is easy to see that $d\mathcal{D}(u)v = - \int v \Delta u$ for any test function $v$ with compact support, implying that the $L^2$-gradient of $\mathcal{D}$ is $-\Delta$ and $\dot{u} = \Delta u$ is the $L^2$-gradient flow of Dirichlet energy: the motion which decreases the quantity $\mathcal{D}(u)$ the fastest *as measured by the $L^2$ norm*. It can be shown that high-frequency modes decay quickly under this flow, while low-frequency information takes much longer to dissipate. On the other hand, we could alternatively run the $H^1$-gradient flow of $\mathcal{D}$, which is motion of fastest decrease with respect to the $H^1$ inner product $(u, v) = \int \langle \nabla u, \nabla v \rangle \, d\mu$. This motion is prescribed in terms of the $H^1$-gradient of $\mathcal{D}$, which by the discussion above with $\mathbf{A}_0 = -\Delta$ is easily seen to be the identity. This means that the $H^1$-gradient flow is given by $\dot{u} = -u$, which retains the minimizers of the $L^2$-flow but with quite different intermediate character, since it functions by simultaneously flattening all spatial frequencies. The process of preconditioning a gradient flow by matching derivatives is known as a Sobolev gradient method (c.f. [69]), and these methods often exhibit faster convergence and better numerical behavior than their $L^2$ counterparts [70].

Returning to the graph setting, our learnable matrices $\mathbf{A}_k$ on $k$-cliques will lead to inner products $(\cdot, \cdot)_k$ on functions in $\Omega_k$, and this will induce dual derivatives as described in Appendix A.1. However, in this case we will not have $d_0^* = d_0^{\mathsf{T}}$, but instead the expression given by the following result:

**Proposition A.3.** *The $\mathbf{A}_k$-adjoints $d_k^*$ to the graph derivative operators $d_k$ are given by $d_k^* = \mathbf{A}_k^{-1} d_k^{\mathsf{T}} \mathbf{A}_{k+1}$. Similarly, for any linear operator $\mathbf{B} : \Omega_k \to \Omega_k$, the $\mathbf{A}_k$-adjoint $\mathbf{B}^* = \mathbf{A}_k^{-1} \mathbf{B}^{\mathsf{T}} \mathbf{A}$.*

*Proof.* Let $\mathbf{q}, \mathbf{p}$ denote vectors of $k$-clique resp. $(k + 1)$-clique features. It follows that

$$(d_k \mathbf{q}, \mathbf{p})_{k+1} = \langle d_k \mathbf{q}, \mathbf{A}_{k+1} \mathbf{p} \rangle = \langle \mathbf{q}, d_k^{\mathsf{T}} \mathbf{A}_{k+1} \mathbf{p} \rangle = \langle \mathbf{q}, \mathbf{A}_k d_k^* \mathbf{p} \rangle = (\mathbf{q}, d_k^* \mathbf{p})_k .$$

Therefore, we see that $d_k^{\mathsf{T}} \mathbf{A}_{k+1} = \mathbf{A}_k d_k^*$ and hence $d_k^* = \mathbf{A}_k^{-1} d_k^{\mathsf{T}} \mathbf{A}_{k+1}$. Similarly, if $\mathbf{q}, \mathbf{q}'$ denote vectors of $k$-clique features, it follows from the $\ell^2$-self-adjointness of $\mathbf{A}_k$ that

$$\left( \mathbf{q}, \mathbf{B} \mathbf{q}' \right)_k = \left\langle \mathbf{q}, \mathbf{A}_k \mathbf{B} \mathbf{q}' \right\rangle = \langle \mathbf{B}^{\mathsf{T}} \mathbf{A}_k \mathbf{q}, \mathbf{q}' \rangle = \left\langle \mathbf{A}_k^{-1} \mathbf{B}^{\mathsf{T}} \mathbf{A}_k \mathbf{q}, \mathbf{A}_k \mathbf{q}' \right\rangle = (\mathbf{B}^* \mathbf{q}, \mathbf{q}')_k ,$$

establishing that $\mathbf{B}^* = \mathbf{A}_k^{-1} \mathbf{B}^{\mathsf{T}} \mathbf{A}_k$. $\qquad\square$

**Remark A.6.** *It is common in graph theory to encounter the case where $a_i > 0$ are nodal weights and $w_{ij} > 0$ are edge weights. These are nothing more than the (diagonal) inner products $\mathbf{A}_0, \mathbf{A}_1$ in disguise, and so Proposition A.3 immediately yields the familiar formula for the induced divergence*

$$(d_0^* \mathbf{p})_i = \frac{1}{a_i} \sum_{j:(i,j) \in \mathcal{E}} w_{ij} \left( \mathbf{p}_{ji} - \mathbf{p}_{ij} \right) .$$

Note that all of these notions extend to the case of block inner products in the obvious way. For example, if $\mathbf{q}, \mathbf{p}$ are node resp. edge features, it follows that $\mathbf{A} = \mathrm{diag}\left( \mathbf{A}_0, \mathbf{A}_1 \right)$ is an inner product on node-edge feature pairs, and the adjoints of node-edge operators with respect to $\mathbf{A}$ are computed as according to Proposition A.3.

**Remark A.7.** *For convenience, this work restricts to diagonal matrices $\mathbf{A}_0, \mathbf{A}_1$. However, note that a matrix which is diagonal in "edge space" $\mathcal{G}_2$ is generally full in a nodal representation. This is because an (undirected) edge is uniquely specified by the two nodes which it connects, meaning that a purely local quantity on edges is necessarily nonlocal on nodes.*

## A.4 Higher order attention

As mentioned in the body, when $f = \exp$ and $\tilde{a}(\mathbf{q}_i, \mathbf{q}_j) = (1/d) \langle \mathbf{W}_K \mathbf{q}_i, \mathbf{W}_Q \mathbf{q}_j \rangle$, defining the learnable inner products $\mathbf{A}_0 = (a_{0,ii})$, $\mathbf{A}_1 = (a_{1,ij})$ as

$$a_{0,ii} = \sum_{j \in \mathcal{N}(i)} f\left(\tilde{a}\left(\mathbf{q}_i, \mathbf{q}_j\right)\right), \qquad a_{1,ij} = f\left(\tilde{a}\left(\mathbf{q}_i, \mathbf{q}_j\right)\right),$$

recovers scaled dot product attention as $\mathbf{A}_0^{-1} \mathbf{A}_1$.

**Remark A.8.** *Technically, $\mathbf{A}_1$ is an inner product only with respect to a predefined ordering of the edges $\alpha = (i, j)$, since we do not require $\mathbf{A}_1$ be orientation-invariant. On the other hand, it is both unnecessary and distracting to enforce symmetry on $\mathbf{A}_1$ in this context, since any necessary symmetrization will be handled automatically by the differential operator $d_0^*$.*

Similarly, other common attention mechanisms are produced by modifying the pre-attention function $\tilde{a}$. While $\mathbf{A}_0^{-1} \mathbf{A}_1$ never appears in the brackets of Section 4, letting $\alpha = (i, j)$ denote a global edge with endpoints $i, j$, it is straightforward to calculate the divergence of an antisymmetric edge feature $\mathbf{p}$ at node $i$,

$$\left(d_0^* \mathbf{p}\right)_i = \left(\mathbf{A}_0^{-1} d_0^\mathsf{T} \mathbf{A}_1 \mathbf{p}\right)_i = a_{0,ii}^{-1} \sum_{\alpha} (d_0^\mathsf{T})_{i\alpha} \left(\mathbf{A}_1 \mathbf{p}\right)_\alpha$$

$$= a_{0,ii}^{-1} \sum_{\alpha \ni i} \left(\mathbf{A}_1 \mathbf{p}\right)_{-\alpha} - \left(\mathbf{A}_1 \mathbf{p}\right)_\alpha = - \sum_{j \in \mathcal{N}(i)} \frac{a_{1,ji} + a_{i,ij}}{a_{0,ii}} \mathbf{p}_{ij}.$$

This shows that $b(\mathbf{q}_i, \mathbf{q}_j) = \left(a_{1,ij} + a_{1,ji}\right)/a_{0,ii}$ appears under the divergence in $d_0^* = \mathbf{A}_0^{-1} d_0^\mathsf{T} \mathbf{A}_1$, which is the usual graph attention up to a symmetrization in $\mathbf{A}_1$.

**Remark A.9.** *While $\mathbf{A}_1$ is diagonal on global edges $\alpha = (i, j)$, it appears sparse nondiagonal in its nodal representation. Similarly, any diagonal extension $\mathbf{A}_2$ to 2-cliques will appear as a sparse 3-tensor $\mathbf{A}_2 = (a_{2,ijk})$ when specified by its nodes.*

This inspires a straightforward extension of graph attention to higher-order cliques. In particular, denote by $K > 0$ the highest degree of clique under consideration, and define $\mathbf{A}_{K-1} = (a_{K-1, i_1 i_2 \ldots i_K})$ by

$$a_{K-1, i_1 i_2 \ldots i_K} = f\left(\mathbf{W}\left(\mathbf{q}_{i_1}, \mathbf{q}_{i_2}, \ldots, \mathbf{q}_{i_K}\right)\right),$$

where $\mathbf{W} \in \mathbb{R}^{\otimes_K n_V}$ is a learnable $K$-tensor. Then, for any $0 \le k \le K - 2$ define $\mathbf{A}_k = \left(a_{k, i_1 i_2 \ldots i_{k+1}}\right)$ by

$$a_{k, i_1 i_2 \ldots i_{k+1}} = \sum_{i_K, \ldots, i_{K-k-1}} a_{K-1, i_1 i_2 \ldots i_K}.$$

This recovers the matrices $\mathbf{A}_0$, $\mathbf{A}_1$ from before when $K = 2$, and otherwise extends the same core idea to higher-order cliques. It's attractive that the attention mechanism captured by $d_k^*$ remains asymmetric, meaning that the attention of any one node to the others in a $k$-clique need not equal the attention of the others to that particular node.

**Remark A.10.** *A more obvious but less expressive option for higher-order attention is to let*

$$a_{k, i_1 i_2 \ldots i_{k+1}} = \frac{a_{K-1, i_1 i_2 \ldots i_K}}{\sum_{i_K, \ldots, i_{K-k-1}} a_{K-1, i_1 i_2 \ldots i_K}},$$

*for any $0 \le k \le K - 2$. However, application of the combinatorial codifferential $d_{k-1}^\mathsf{T}$ appearing in $d_{k-1}^*$ will necessarily symmetrize this quantity, so that the asymmetry behind the attention mechanism is lost in this formulation.*

To illustrate how this works more concretely, consider the extension $K = 3$ to 2-cliques, and let $\mathcal{N}(i, j) = \mathcal{N}(i) \cap \mathcal{N}(j)$. We have the tensors $\mathbf{A}_2 = (a_{2,ijk})$, $\mathbf{A}_1 = (a_{1,ij})$, and $\mathbf{A}_0 = (a_{0,i})$ defined by

$$a_{2,ijk} = f\left(\mathbf{W}\left(\mathbf{q}_i, \mathbf{q}_j, \mathbf{q}_k\right)\right), \quad a_{1,ij} = \sum_{k \in \mathcal{N}(i,j)} a_{2,ijk}, \quad a_{0,i} = \sum_{j \in \mathcal{N}(i)} \sum_{k \in \mathcal{N}(i,j)} a_{2,ijk}.$$

This provides a way for (features on) 3-node subgraphs of $\mathcal{G}$ to attend to each other, and can be similarly built-in to the differential operator $d_1^* = \mathbf{A}_1^{-1} d_0^\mathsf{T} \mathbf{A}_2$.

## A.5 Exterior calculus interpretation of GATs

Let $\mathcal{N}(i)$ denote the one-hop neighborhood of node $i$, and let $\overline{\mathcal{N}}(i) = \mathcal{N}(i) \cup \{i\}$. Recall the standard (single-headed) graph attention network (GAT) described in [12], described layer-wise as

$$\mathbf{q}_i^{k+1} = \sigma\left(\sum_{j \in \overline{\mathcal{N}}(i)} a\left(\mathbf{q}_i^k, \mathbf{q}_j^k\right) \mathbf{W}^k \mathbf{q}_j^k\right), \tag{1}$$

where $\sigma$ is an element-wise nonlinearity, $\mathbf{W}^k$ is a layer-dependent embedding matrix, and $a\left(\mathbf{q}_i, \mathbf{q}_j\right)$ denotes the attention node $i$ pays to node $j$. Traditionally, the attention mechanism is computed through

$$a\left(\mathbf{q}_i, \mathbf{q}_j\right) = \mathrm{Softmax}_j \, \tilde{a}\left(\mathbf{q}_i, \mathbf{q}_j\right) = \frac{e^{\tilde{a}(\mathbf{q}_i, \mathbf{q}_j)}}{\sigma_i},$$

where the pre-attention coefficients $\tilde{a}\left(\mathbf{q}_i, \mathbf{q}_j\right)$ and nodal weights $\sigma_i$ are defined as

$$\tilde{a}\left(\mathbf{q}_i, \mathbf{q}_j\right) = \mathrm{LeakyReLU}\left(\mathbf{a}^{\intercal}\left(\mathbf{W}^{\intercal}\mathbf{q}_i \,\|\, \mathbf{W}^{\intercal}\mathbf{q}_j\right)\right), \qquad \sigma_i = \sum_{j \in \overline{N}(i)} e^{\tilde{a}(\mathbf{q}_i, \mathbf{q}_j)}.$$

However, the exponentials in the outer $\mathrm{Softmax}$ are often replaced with other nonlinear functions, e.g. Squareplus, and the pre-attention coefficients $\tilde{a}$ appear as variable (but learnable) functions of the nodal features. First, notice that (1) the attention coefficients $a\left(\mathbf{q}_i, \mathbf{q}_j\right)$ depend on the node features $\mathbf{q}$ and not simply the topology of the graph, and (2) the attention coefficients are not symmetric, reflecting the fact that the attention paid by node $i$ to node $j$ need not equal the attention paid by node $j$ to node $i$. A direct consequence of this is that GATs are not purely diffusive under any circumstances, since it was shown in Appendix A.1 that the combinatorial divergence $d_0^{\intercal}$ will antisymmetrize the edge features it acts on. In particular, it is clear that the product $a\left(\mathbf{q}_i, \mathbf{q}_j\right)\left(\mathbf{q}_i - \mathbf{q}_j\right)$ is asymmetric in $i, j$ under the standard attention mechanism, since even the pre-attention coefficients $\tilde{a}\left(\mathbf{q}_i, \mathbf{q}_j\right)$ are not symmetric, meaning that there will be two distinct terms after application of the divergence. More precisely, there is the following subtle result.

**Proposition A.4.** *Let* $\mathbf{q} \in \mathbb{R}^{|\mathcal{V}| \times n_V}$ *denote an array of nodal features. The expression*

$$\sum_{j \in \overline{\mathcal{N}}(i)} a\left(\mathbf{q}_i, \mathbf{q}_j\right)\left(\mathbf{q}_i - \mathbf{q}_j\right),$$

*where* $a = \mathbf{A}_0^{-1}\mathbf{A}_1$ *is not the action of a Laplace operator whenever* $\mathbf{A}_1$ *is not symmetric.*

*Proof.* From Appendix A.3, we know that any Laplace operator on nodes is expressible as $d_0^* d_0 = \mathbf{A}_0^{-1} d_0^{\intercal} \mathbf{A}_1 d_0$ for some positive definite $\mathbf{A}_0, \mathbf{A}_1$. So, we compute the action of the Laplacian at node $i$,

$$(\Delta_0 \mathbf{q})_i = (d_0^* d_0 \mathbf{q})_i = \left(\mathbf{A}_0^{-1} d_0^{\intercal} \mathbf{A}_1 d_0 \mathbf{q}\right)_i = a_{0,ii}^{-1} \sum_{\alpha} (d_0^{\intercal})_{i\alpha}\left(\mathbf{A}_1 d_0 \mathbf{q}\right)_{\alpha}$$

$$= a_{0,ii}^{-1} \sum_{\alpha \ni i}\left(\mathbf{A}_1 d_0 \mathbf{q}\right)_{-\alpha} - \left(\mathbf{A}_1 d_0 \mathbf{q}\right)_{\alpha} = -\frac{1}{2}\sum_{j \in \mathcal{N}(i)} \frac{a_{1,ji} + a_{i,ij}}{a_{0,ii}}\left(\mathbf{q}_j - \mathbf{q}_i\right),$$

$$= \sum_{j \in \mathcal{N}(i)} a\left(\mathbf{q}_i, \mathbf{q}_j\right)\left(\mathbf{q}_j - \mathbf{q}_i\right),$$

which shows that $a\left(\mathbf{q}_i, \mathbf{q}_j\right) = (1/2)\left(a_{1,ji} + a_{1,ij}\right)/a_{0,ii}$ must have symmetric numerator. $\square$

While this result shows that GATs (and their derivatives, e.g., GRAND) are not purely diffusive, it also shows that it is possible to get close to GAT (at least syntactically) with a learnable diffusion mechanism. In fact, setting $\sigma = \mathbf{W}^k = \mathbf{I}$ in (1) yields precisely a single-step diffusion equation provided that $a\left(q_i^k, q_j^k\right)$ is right-stochastic (i.e., $\sum_j a\left(\mathbf{q}_i, \mathbf{q}_j\right) 1_j = 1_i$) and built as dictated by Proposition A.4.

**Theorem A.11.** *The GAT layer* (1) *is a single-step diffusion equation provided that* $\sigma = \mathbf{W}^k = \mathbf{I}$, *and the attention mechanism* $a\left(\mathbf{q}_i, \mathbf{q}_j\right) = (1/2)\left(a_{1,ji} + a_{1,ij}\right)/a_{0,ii}$ *is right-stochastic.*

*Proof.* First, notice that the Laplacian with respect to an edge set which contains self-loops is computable via

$$(\Delta_0 \mathbf{q})_i = - \sum_{j \in \overline{\mathcal{N}}(i)} a\left(\mathbf{q}_i, \mathbf{q}_j\right)\left(\mathbf{q}_j - \mathbf{q}_i\right) = \mathbf{q}_i - \sum_{j \in \overline{\mathcal{N}}(i)} a\left(\mathbf{q}_i, \mathbf{q}_j\right) \mathbf{q}_j.$$

Therefore, taking a single step of heat flow $\dot{\mathbf{q}} = -\Delta_0 \mathbf{q}$ with forward Euler discretization and time step $\tau = 1$ is equivalent to

$$\mathbf{q}_i^{k+1} = \mathbf{q}_i^k - \tau \left(\Delta_0 \mathbf{q}^k\right)_i = \sum_{j \in \overline{\mathcal{N}}(i)} a\left(\mathbf{q}_i^k, \mathbf{q}_j^k\right) \mathbf{q}_j^k,$$

which is just a modified and non-activated GAT layer with $\mathbf{W}^k = \mathbf{I}$ and attention mechanism $a$. $\quad\square$

**Remark A.12.** *Since* Softmax *and its variants are right-stochastic, Theorem A.11 is what establishes equivalence between the non-divergence equation*

$$\dot{\mathbf{q}}_i = \sum_{j \in \mathcal{N}(i)} a\left(\mathbf{q}_i, \mathbf{q}_j\right)\left(\mathbf{q}_j - \mathbf{q}_i\right),$$

*and the standard GAT layer seen in, e.g., [49], when $a(\mathbf{q}_i, \mathbf{q}_j)$ is the usual attention mechanism.*

**Remark A.13.** *In the literature, there is an important (and often overlooked) distinction between the positive graph/Hodge Laplacian $\Delta_0$ and the negative "geometer's Laplacian" $\Delta$ which is worth noting here. Particularly, we have from integration-by-parts that the gradient $\nabla = d_0$ is $L^2$-adjoint to minus the divergence $-\nabla \cdot = d_0^\mathsf{T}$, so that the two Laplace operators $\Delta_0 = d_0^\mathsf{T} d_0$ and $\Delta = \nabla \cdot \nabla$ differ by a sign. This is why the same $\ell^2$-gradient flow of Dirichlet energy can be equivalently expressed as $\dot{\mathbf{q}} = \Delta \mathbf{q} = -\Delta_0 \mathbf{q}$, but not by, e.g., $\dot{\mathbf{q}} = \Delta_0 \mathbf{q}$.*

This shows that, while they are not equivalent, there is a close relationship between attention and diffusion mechanisms on graphs. The closest analogue to the standard attention expressible in this format is perhaps the choice $a_{1,ij} = f\left(\tilde{a}\left(\mathbf{q}_i, \mathbf{q}_j\right)\right)$, $a_{0,ii} = \sum_{j \in \bar{\mathcal{N}}(i)} a_{1,ij}$, discussed in Section 4 and Appendix A.4, where $f$ is any scalar-valued positive function. For example, when $f(x) = e^x$, it follows that

$$(\Delta_0 \mathbf{q})_i = -\frac{1}{2} \sum_{j \in \mathcal{N}(i)} \frac{e^{\tilde{a}(\mathbf{q}_i, \mathbf{q}_j)} + e^{\tilde{a}(\mathbf{q}_j, \mathbf{q}_i)}}{\sigma_i} \left(\mathbf{q}_j - \mathbf{q}_i\right)$$

$$= -\frac{1}{2} \sum_{j \in \mathcal{N}(i)} \left( a\left(\mathbf{q}_i, \mathbf{q}_j\right) + \frac{e^{\tilde{a}(\mathbf{q}_j, \mathbf{q}_i)}}{\sigma_i} \right) \left(\mathbf{q}_j - \mathbf{q}_i\right),$$

which leads to the standard GAT propagation mechanism plus an extra term arising from the fact that the attention $a$ is not symmetric.

**Remark A.14.** *Practically, GATs and their variants typically make use of multi-head attention, defined in terms of an attention mechanism which is averaged over some number $|h|$ of independent "heads",*

$$a\left(\mathbf{q}_i, \mathbf{q}_j\right) = \frac{1}{|h|} \sum_h a^h\left(\mathbf{q}_i, \mathbf{q}_j\right),$$

*which are distinct only in their learnable parameters. While the results of this section were presented in terms of $|h| = 1$, the reader can check that multiple attention heads can be used in this framework provided it is the pre-attention $\tilde{a}$ that is averaged instead.*

## A.6   Bracket derivations and properties

Here the architectures in the body are derived in greater detail. First, it will be shown that $\mathbf{L}^* = -\mathbf{L}$, $\mathbf{G}^* = \mathbf{G}$, and $\mathbf{M}^* = \mathbf{M}$, as required for structure-preservation.

**Proposition A.5.** *For $\mathbf{L}, \mathbf{G}, \mathbf{M}$ defined in Section 4, we have $\mathbf{L}^* = -\mathbf{L}$, $\mathbf{G}^* = \mathbf{G}$, and $\mathbf{M}^* = \mathbf{M}$.*

*Proof.* First, denoting $\mathbf{A} = \mathrm{diag}\,(\mathbf{A}_0, \mathbf{A}_1)$, it was shown in section A.3 that $\mathbf{B}^* = \mathbf{A}^{-1}\mathbf{B}^\mathsf{T}\mathbf{A}$ for any linear operator $\mathbf{B}$ of appropriate dimensions. So, applying this to $\mathbf{L}$, it follows that

$$\mathbf{L}^* = \begin{pmatrix} \mathbf{A}_0^{-1} & \mathbf{0} \\ \mathbf{0} & \mathbf{A}_1^{-1} \end{pmatrix} \begin{pmatrix} \mathbf{0} & -d_0^* \\ d_0 & \mathbf{0} \end{pmatrix}^\mathsf{T} \begin{pmatrix} \mathbf{A}_0 & \mathbf{0} \\ \mathbf{0} & \mathbf{A}_1 \end{pmatrix}$$

$$= \begin{pmatrix} \mathbf{0} & \mathbf{A}_0^{-1}d_0^\mathsf{T}\mathbf{A}_1 \\ -\mathbf{A}_1^{-1}(d_0^*)^\mathsf{T}\mathbf{A}_0 & \mathbf{0} \end{pmatrix} = \begin{pmatrix} \mathbf{0} & d_0^* \\ -d_0 & \mathbf{0} \end{pmatrix} = -\mathbf{L}.$$

Similarly, it follows that

$$\mathbf{G}^* = \begin{pmatrix} \mathbf{A}_0^{-1} & \mathbf{0} \\ \mathbf{0} & \mathbf{A}_1^{-1} \end{pmatrix} \begin{pmatrix} d_0^*d_0 & \mathbf{0} \\ \mathbf{0} & d_1^*d_1 \end{pmatrix}^\mathsf{T} \begin{pmatrix} \mathbf{A}_0 & \mathbf{0} \\ \mathbf{0} & \mathbf{A}_1 \end{pmatrix}$$

$$= \begin{pmatrix} \mathbf{A}_0^{-1}d_0^\mathsf{T}(d_0^*)^\mathsf{T}\mathbf{A}_0 & \mathbf{0} \\ \mathbf{0} & \mathbf{A}_1^{-1}d_1^\mathsf{T}(d_1^*)^\mathsf{T}\mathbf{A}_1 \end{pmatrix} = \begin{pmatrix} d_0^*d_0 & \mathbf{0} \\ \mathbf{0} & d_1^*d_1 \end{pmatrix} = \mathbf{G},$$

$$\mathbf{M}^* = \begin{pmatrix} \mathbf{A}_0^{-1} & \mathbf{0} \\ \mathbf{0} & \mathbf{A}_1^{-1} \end{pmatrix} \begin{pmatrix} \mathbf{0} & \mathbf{0} \\ \mathbf{0} & \mathbf{A}_1 d_1^* d_1 \mathbf{A}_1 \end{pmatrix}^\mathsf{T} \begin{pmatrix} \mathbf{A}_0 & \mathbf{0} \\ \mathbf{0} & \mathbf{A}_1 \end{pmatrix}$$

$$= \begin{pmatrix} \mathbf{0} & \mathbf{0} \\ \mathbf{0} & d_1^\mathsf{T}(d_1^*)^\mathsf{T}\mathbf{A}_1^2 \end{pmatrix} = \begin{pmatrix} \mathbf{0} & \mathbf{0} \\ \mathbf{0} & \mathbf{A}_1 d_1^* d_1 \mathbf{A}_1 \end{pmatrix} = \mathbf{M},$$

where the second-to-last equality used that $\mathbf{A}_1\mathbf{A}_1^{-1} = \mathbf{I}$. □

**Remark A.15.** *Note that the choice of zero blocks in $\mathbf{L}, \mathbf{G}$ is sufficient but not necessary for these adjointness relationships to hold. For example, one could alternatively choose the diagonal blocks of $\mathbf{L}$ to contain terms like $\mathbf{B} - \mathbf{B}^*$ for an appropriate message-passing network $\mathbf{B}$.*

Next, we compute the gradients of energy and entropy with respect to $(\cdot, \cdot)$.

**Proposition A.6.** *The $\mathbf{A}$-gradient of the energy*

$$E(\mathbf{q}, \mathbf{p}) = \frac{1}{2}\left(|\mathbf{q}|^2 + |\mathbf{p}|^2\right) = \frac{1}{2}\sum_{i\in\mathcal{V}}|\mathbf{q}_i|^2 + \frac{1}{2}\sum_{\alpha\in\mathcal{E}}|\mathbf{p}_\alpha|^2,$$

*satisfies*

$$\nabla E(\mathbf{q}, \mathbf{p}) = \begin{pmatrix} \mathbf{A}_0^{-1} & \mathbf{0} \\ \mathbf{0} & \mathbf{A}_1^{-1} \end{pmatrix}\begin{pmatrix} \mathbf{q} \\ \mathbf{p} \end{pmatrix} = \begin{pmatrix} \mathbf{A}_0^{-1}\mathbf{q} \\ \mathbf{A}_1^{-1}\mathbf{p} \end{pmatrix}.$$

*Moreover, given the energy and entropy defined as*

$$E(\mathbf{q}, \mathbf{p}) = f_E\left(s(\mathbf{q})\right) + g_E\left(s\left(d_0 d_0^\mathsf{T}\mathbf{p}\right)\right),$$
$$S(\mathbf{q}, \mathbf{p}) = g_S\left(s\left(d_1^\mathsf{T} d_1 \mathbf{p}\right)\right),$$

*where $f_E : \mathbb{R}^{n_\mathcal{V}} \to \mathbb{R}$ acts on node features, $g_E, g_S : \mathbb{R}^{n_\mathcal{E}} \to \mathbb{R}$ act on edge features, and $s$ denotes sum aggregation over nodes or edges, the $\mathbf{A}$-gradients are*

$$\nabla E(\mathbf{q}, \mathbf{p}) = \begin{pmatrix} \mathbf{A}_0^{-1}\mathbf{1} \otimes \nabla f_E\left(s(\mathbf{q})\right) \\ \mathbf{A}_1^{-1}d_0 d_0^\mathsf{T}\mathbf{1} \otimes \nabla g_E\left(s\left(d_0 d_0^\mathsf{T}\mathbf{p}\right)\right) \end{pmatrix}, \quad \nabla S(\mathbf{q}, \mathbf{p}) = \begin{pmatrix} \mathbf{0} \\ \mathbf{A}_1^{-1}d_1^\mathsf{T} d_1 \mathbf{1} \otimes \nabla g_S\left(s\left(d_1^\mathsf{T} d_1 \mathbf{p}\right)\right) \end{pmatrix},$$

*Proof.* Since the theory of $\mathbf{A}$-gradients in section A.3 establishes that $\nabla E = \mathbf{A}^{-1}\delta E$, it is only necessary to compute the $L^2$-gradients. First, letting $\mathbf{x} = (\mathbf{q} \quad \mathbf{p})^\mathsf{T}$, it follows for the first definition of energy that

$$dE(\mathbf{x}) = \sum_{i\in\mathcal{V}}\langle\mathbf{q}_i, d\mathbf{q}_i\rangle + \sum_{\alpha\in\mathcal{E}}\langle\mathbf{p}_\alpha, d\mathbf{p}_\alpha\rangle = \langle\mathbf{q}, d\mathbf{q}\rangle + \langle\mathbf{p}, d\mathbf{p}\rangle = \langle\mathbf{x}, d\mathbf{x}\rangle,$$

showing that $\delta E(\mathbf{q}, \mathbf{p}) = (\mathbf{q} \quad \mathbf{p})^\mathsf{T}$, as desired. Moving to the metriplectic definitions, since each term of $E, S$ has the same functional form, it suffices to compute the gradient of $f\left(s\left(\mathbf{B}\mathbf{q}\right)\right)$ for some function $f : \mathbb{R}^{n_f} \to \mathbb{R}$ and matrix $\mathbf{B} : \mathbb{R}^{|\mathcal{V}|} \to \mathbb{R}^{|\mathcal{V}|}$. To that end, adopting the Einstein summation convention where repeated indices appearing up-and-down in an expression are implicitly summed, if $1 \le a, b \le n_f$ and $1 \le i, j \le |\mathcal{V}|$, we have

$$d\left(s(\mathbf{q})\right) = \sum_{i\in|\mathcal{V}|} d\mathbf{q}_i = \sum_{i\in|\mathcal{V}|} \delta_i^j d\mathbf{q}_j = \mathbf{1}^j dq_j^a \mathbf{e}_a = (\mathbf{1} \otimes \mathbf{I}) : d\mathbf{q} = \nabla\left(s(\mathbf{q})\right) : d\mathbf{q},$$

implying that $\nabla s(\mathbf{q}) = \mathbf{1} \otimes \mathbf{I}$. Continuing, it follows that

$$d\left(f \circ s \circ \mathbf{B}\mathbf{q}\right) = f'\left(s\left(\mathbf{B}\mathbf{q}\right)\right)_a s'\left(\mathbf{B}\mathbf{q}\right)_i^a B^{ij} dq_j^a = f'\left(s\left(\mathbf{B}\mathbf{q}\right)\right)_a \mathbf{e}_a \left(B^\intercal\right)^{ij} 1_j \, dq_i^a$$
$$= \langle \nabla f\left(s\left(\mathbf{B}\mathbf{q}\right)\right) \otimes \mathbf{B}^\intercal \mathbf{1}, d\mathbf{q}\rangle = \langle \nabla\left(f \circ s \circ \mathbf{B}\mathbf{q}\right), d\mathbf{q}\rangle,$$

showing that $\nabla\left(f \circ s \circ \mathbf{B}\right)$ decomposes into an outer product across modalities. Applying this formula to the each term of $E, S$ then yields the $L^2$-gradients,

$$\delta E(\mathbf{q}, \mathbf{p}) = \begin{pmatrix} \mathbf{1} \otimes \nabla f_E\left(s(\mathbf{q})\right) \\ d_0 d_0^\intercal \mathbf{1} \otimes \nabla g_E\left(s\left(d_0 d_0^\intercal \mathbf{p}\right)\right) \end{pmatrix}, \quad \delta S(\mathbf{q}, \mathbf{p}) = \begin{pmatrix} \mathbf{0} \\ d_1^\intercal d_1 \mathbf{1} \otimes \nabla g_S\left(s\left(d_1^\intercal d_1 \mathbf{p}\right)\right) \end{pmatrix},$$

from which the desired $\mathbf{A}$-gradients follow directly. $\qquad\square$

Finally, we can show that the degeneracy conditions for metriplectic structure are satisfied by the network in Section 4.

**Theorem A.16.** *The degeneracy conditions* $\mathbf{L}\nabla S = \mathbf{M}\nabla E = \mathbf{0}$ *are satisfied by the metriplectic bracket in Section A.6.*

*Proof.* This is a direct calculation using Theorem 3.1 and Proposition A.6. In particular, it follows that

$$\mathbf{L}\nabla S = \begin{pmatrix} \mathbf{0} & -d_0^* \\ d_0 & \mathbf{0} \end{pmatrix} \begin{pmatrix} \mathbf{0} \\ \mathbf{A}_1^{-1} d_1^\intercal d_1 \mathbf{1} \otimes \nabla g_S\left(s\left(d_1^\intercal d_1 \mathbf{p}\right)\right) \end{pmatrix}$$
$$= \begin{pmatrix} -\mathbf{A}_0^{-1}\left(d_1 d_0\right)^\intercal d_1 \mathbf{1} \otimes \nabla g_S\left(s\left(d_1^\intercal d_1 \mathbf{p}\right)\right) \\ \mathbf{0} \end{pmatrix} = \begin{pmatrix} \mathbf{0} \\ \mathbf{0} \end{pmatrix},$$

$$\mathbf{M}\nabla E = \begin{pmatrix} \mathbf{0} & \mathbf{0} \\ \mathbf{0} & \mathbf{A}_1 d_1^* d_1 \mathbf{A}_1 \end{pmatrix} \begin{pmatrix} \mathbf{A}_0^{-1} \mathbf{1} \otimes \nabla f_E\left(s(\mathbf{q})\right) \\ \mathbf{A}_1^{-1} d_0 d_0^\intercal \mathbf{1} \otimes \nabla g_E\left(s\left(d_0 d_0^\intercal \mathbf{p}\right)\right) \end{pmatrix}$$
$$= \begin{pmatrix} \mathbf{0} \\ \mathbf{A}_1 d_1^*\left(d_1 d_0\right) d_0^\intercal \mathbf{1} \otimes \nabla g_E\left(s\left(d_0 d_0^\intercal \mathbf{p}\right)\right) \end{pmatrix} = \begin{pmatrix} \mathbf{0} \\ \mathbf{0} \end{pmatrix},$$

since $d_1 d_0 = 0$ as a consequence of the graph calculus. These calculations establish the validity of the energy conservation and entropy generation properties seen previously in the manuscript body. $\quad\square$

**Remark A.17.** *Clearly, this is not the only possible metriplectic formulation for GNNs. On the other hand, this choice is in some sense maximally general with respect to the chosen operators* $\mathbf{L}, \mathbf{G}$, *since only constants are in the kernel of* $d_0$ *(hence there is no reason to include a nodal term in $S$), and only elements in the image of* $d_1^\intercal$ *(which do not exist in our setting) are guaranteed to be in the kernel of* $d_0^\intercal$ *for any graph. Therefore,* $\mathbf{M}$ *is chosen to be essentially* $\mathbf{G}$ *without the* $\Delta_0$ *term, whose kernel is graph-dependent and hence difficult to design.*

# B  Experimental details and more results

This Appendix provides details regarding the experiments in Section 5, as well as any additional information necessary for reproducing them. We implement the proposed algorithms with PYTHON and PYTORCH [71] that supports CUDA. The experiments are conducted on systems that are equipped with NVIDIA RTX A100 and V100 GPUs. For NODEs capabilities, we use the TORCHDIFFEQ library [16].

## B.1  Damped double pendulum

The governing equations for the damped double pendulum can be written in terms of four coupled first-order ODEs for the angles that the two pendula make with the vertical axis $\theta_1, \theta_2$ and their associated angular momenta $\omega_1, \omega_2$ (see [72]),

$$\dot{\theta}_i = \omega_i, \qquad 1 \le i \le 2, \tag{2}$$

$$\dot{\omega}_1 = \frac{m_2 l_1 \omega_1^2 \sin\left(2\Delta\theta\right) + 2 m_2 l_2 \omega_2^2 \sin\left(\Delta\theta\right) + 2 g m_2 \cos\theta_2 \sin\Delta\theta + 2 g m_1 \sin\theta_1 + \gamma_1}{-2 l_1 \left(m_1 + m_2 \sin^2 \Delta\theta\right)}, \tag{3}$$

$$\dot{\omega}_2 = \frac{m_2 l_2 \omega_2^2 \sin\left(2\Delta\theta\right) + 2\left(m_1 + m_2\right) l_1 \omega_1^2 \sin\Delta\theta + 2 g\left(m_1 + m_2\right)\cos\theta_1 \sin\Delta\theta + \gamma_2}{2 l_2 \left(m_1 + m_2 \sin^2 \Delta\theta\right)}, \tag{4}$$

where $m_1, m_2, l_1, l_2$ are the masses resp. lengths of the pendula, $\Delta\theta = \theta_1 - \theta_2$ is the (signed) difference in vertical angle, $g$ is the acceleration due to gravity, and

$$\gamma_1 = 2k_1\dot{\theta}_1 - 2k_2\dot{\theta}_2\cos\Delta\theta.$$

$$\gamma_2 = 2k_1\dot{\theta}_1\cos\Delta\theta - \frac{2(m_1 + m_2)}{m_2}k_2\dot{\theta}_2,$$

for damping constants $k_1, k_2$.

**Dataset.** A trajectory of the damped double pendulum by solving an initial value problem associated with the ODE 2. The initial condition used is $(1.0, \pi/2, 0.0, 0.0)$, and the parameters are $m_1 = m_2 = 1, g = 1, l_1 = 1, l_2 = 0.9, k_1 = k_2 = 0.1$. For time integrator, we use the TorchDiffeq library [16] with Dormand–Prince 5 (DOPRI5) as the numerical solver. The total simulation time is 50 (long enough for significant dissipation to occur), and solution snapshots are collected at 500 evenly-spaced temporal indices.

To simulate the practical case where only positional data for the system is available, the double pendulum solution is integrated to time $T = 50$ (long enough for significant dissipation to occur) and post-processed once the angles and angular momenta are determined from the equations above, yielding the $(x, y)$-coordinates of each pendulum mass at intervals of 0.1s. This is accomplished using the relationships

$$x_1 = l_1\sin\theta_1$$
$$y_1 = -l_1\cos\theta_1$$
$$x_2 = x_1 + l_2\sin\theta_2 = l_1\sin\theta_1 + l_2\sin\theta_2$$
$$y_2 = y_1 - l_2\cos\theta_2 = -l_1\cos\theta_1 - l_2\cos\theta_2.$$

The double pendulum is then treated as a fully connected three-node graph with positional coordinates $\mathbf{q}_i = (x_i, y_i)$ as nodal features, and relative velocities $\mathbf{p}_\alpha = (d_0\mathbf{q})_\alpha$ as edge features. Note that the positional coordinates $(x_0, y_0) = (0, 0)$ of the anchor node are held constant during training. To allow for the necessary flexibility of coordinate changes, each architecture from Section 4 makes use of a message-passing feature encoder before time integration, acting on node features and edge features separately, with corresponding decoder returning the original features after time integration.

To elicit a fair comparison, both the NODE and NODE+AE architectures are chosen to contain comparable numbers of parameters to the bracket architectures ($\sim$ 30k), and all networks are trained for 100,000 epochs. For each network, the configuration of weights producing the lowest overall error during training is used for prediction.

**Hyperparameters.** The networks are trained to reconstruct the node/edge features in mean absolute error (MAE) using the Adam optimizer [73]. The NODEs and metriplectic bracket use an initial learning rate of $10^{-4}$, while the other models use an initial learning rate of $10^{-3}$. The width of the hidden layers in the message passing encoder/decoder is 64, and the number of hidden features for nodes/edges is 32. The time integrator used is simple forward Euler.

**Network architectures.** The message passing encoders/decoders are 3-layer MLPs mapping, in the node case, nodal features and their graph derivatives, and in the edge case, edge features and their graph coderivatives, to a hidden representation. For the bracket architectures, the attention mechanism used in the learnable coderivatives is scaled dot product. The metriplectic network uses 2-layer MLPs $f_E, g_E, g_S$ with scalar output and hidden width 64. For the basic NODE, node and edge features are concatenated, flattened, and passed through a 4-layer fully connected network of width 128 in each hidden layer, before being reshaped at the end. The NODE+AE architecture uses a 3-layer fully connected network which operates on the concatenated and flattened latent embedding of size $32 * 6 = 192$, with constant width throughout all layers.

### B.2  Mujoco

We represent an object as a fully-connected graph, where a node corresponds to a body part of the object and, thus, the nodal feature corresponds to a position of a body part or joint. To learn the dynamics of an object, we again follow the encoder-decoder-type architecture considered in the

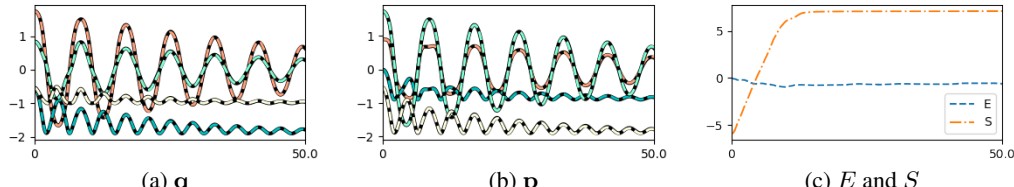

|   |   |   |
|---|---|---|
| (a) **q** | (b) **p** | (c) $E$ and $S$ |

Figure 5: [Double pendulum] Trajectories of **q** and **p**: ground-truth (solid lines) and predictions of the metriplectic bracket model (dashed lines). The evolution of the energy $E$ and the entropy $S$ over the simulation time. Note that slight fluctuations appear in $E$ due to the fact that forward Euler is not a symplectic integrator.

double-pendulum experiments. First we employ a node-wise linear layer to embed the nodal feature into node-wise hidden representations (i.e., the nodal feature $\mathbf{q}_i$ corresponds to a position of a body part or an angle of a joint.). As an alternative encoding scheme for the nodal feature, in addition to the position or the angle, nodal velocities are considered as additional nodal features, i.e., $\mathbf{q}_i = (q_i, v_i)$. The experimental results of the alternative scheme is represented in the following section B.2.2.

The proposed dynamics models also require edge features (e.g., edge velocity), which are not presented in the dataset. Thus, to extract a hidden representation for an edge, we employ a linear layer, which takes velocities of the source and destination nodes of the edge as an input and outputs edge-wise hidden representations, i.e., the edge feature correspond to a pair of nodal velocities $\mathbf{p}_\alpha = (v_{\text{src}(\alpha)}, v_{\text{dst}(\alpha)})$, where $v_{\text{src}(\alpha)}$ and $v_{\text{dst}(\alpha)}$ denote velocities of the source and destination nodes connected to the edge.

The MuJoCo trajectories are generated in the presence of an actor applying controls. To handle the changes in dynamics due to the control input, we introduce an additive forcing term, parameterized by an MLP, to the dynamics models, which is a similar approach considered in dissipative SymODEN [32]. In dissipative SymODEN, the forcing term is designed to affect only the change of the generalized momenta (also known as the port-Hamiltonian dynamics [74]). As opposed to this approach, our proposed forcing term affects the evolution of both the generalized coordinates that are defined in the latent space. Once the latent states are computed at specified time indices, a node-wise linear decoder is applied to reconstruct the position of body parts of the object. Then the models are trained based on the data matching loss measured in mean-square errors between the reconstructed and the ground-truth positions.

### B.2.1 Experiment details

We largely follow the experimental settings considered in [23].

**Dataset.** As elaborated in [23], the standard Open AI Gym [75] environments preprocess observations in ad-hoc ways, e.g., Hopper clips the velocity observations to $[-10, 10]^d$. Thus, the authors in [23] modified the environments to simply return the position and the velocity $(q, v)$ as the observation and we use the same dataset, which is made publicly available by the authors. The dataset consists of training and test data, which are constructed by randomly splitting the episodes in the replay buffer into training and test data. Training and test data consist of $\sim$40K and $\sim$300 or $\sim$ 85 trajectories, respectively. For both training and test data, we include the first 20 measurements (i.e., 19 transitions) in each trajectory.

**Hyperparameters.** For training, we use the Adam optimizer [73] with the initial learning rate 5e-3 and weight decay 1e-4. With the batch size of 200 trajectories, we train the models for 256 epochs. We also employ a cosine annealing learning rate scheduler with the minimum learning rate 1e-6. For time integrator, we use the Torchdiffeq library with the Euler method.

**Network architectures.** The encoder and decoder networks are parameterized as a linear layer and the dimension of the hidden representations is set to 80. For attention, the scaled dot-product attention is used with 8 heads and the embedding dimension is set to 16. The MLP for handling the

forcing term consists of three fully-connected layers (i.e., input, output layers and one hidden layer with 128 neurons). The MLP used for parameterizing the "black-box" NODEs also consists of three fully-connected layers with 128 neurons in each layer.

### B.2.2 Additional results

Figure 6 reports the loss trajectories for all considered dynamics models. For the given number of maximum epochs (i.e., 256), the Hamiltonian and double bracket models tend to reach much lower training losses (an order of magnitude smaller) errors than the NODE and Gradient models do. The metriplectic model produces smaller training losses compared to the NODE and gradient models after a certain number of epochs (e.g., 100 epochs for Hopper).

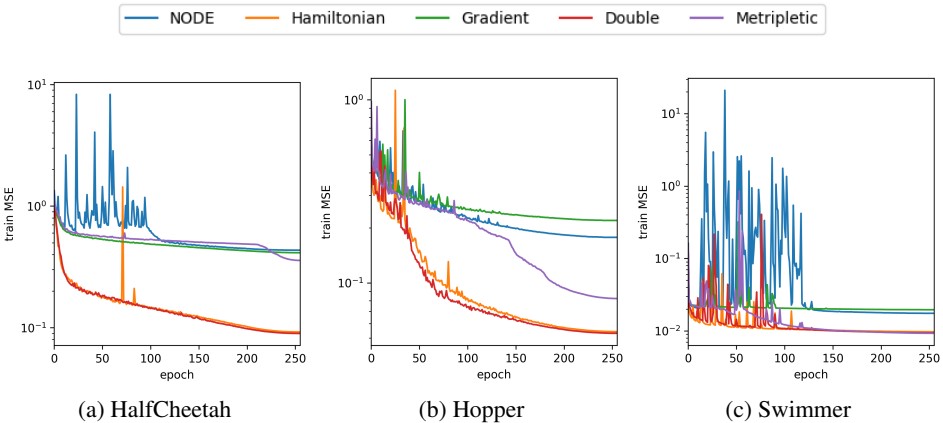

|  | (a) HalfCheetah | (b) Hopper | (c) Swimmer |

Figure 6: [Mujoco] Train MSE over epoch for all considered dynamics models. For the nodal feature, only the position or the angle of the body part/joint is considered.

In the next set of experiments, we provide not only positions/angles of body parts as nodal features, but also velocities of the body parts as nodal features (i.e., $\mathbf{q}_i = (q_i, v_i)$). Table 6 reports the relative errors measured in L2-norm; again, the Hamiltonian, double bracket, and metriplectic outperform other dynamics models. In particular, the metriplectic bracket produces the most accurate predictions in the Hopper and Swimmer environments. Figure 7 reports the loss trajectories for all considered models. Similar to the previous experiments with the position as the only nodal feature, the Hamiltonian and Double bracket produces the lower training losses than the NODE and Gradient models do. For the Hopper and Swimmer environments, however, among all considered models, the metriplectic model produces the lowest training MSEs after 256 training epochs.

| Dataset | HalfCheetah | Hopper | Swimmer |
|---|---|---|---|
| NODE+AE | $0.0848 \pm 0.0011$ | $0.0421 \pm 0.0041$ | $0.0135 \pm 0.00082$ |
| Hamiltonian | $0.0403 \pm 0.0052$ | $0.0294 \pm 0.0028$ | $0.0120 \pm 0.00022$ |
| Gradient | $0.0846 \pm 0.00358$ | $0.0490 \pm 0.0013$ | $0.0158 \pm 0.00030$ |
| Double Bracket | $0.0653 \pm 0.010$ | $0.0274 \pm 0.00090$ | $0.0120 \pm 0.00060$ |
| Metriplectic | $0.0757 \pm 0.0021$ | $0.0269 \pm 0.00035$ | $0.0114 \pm 0.00067$ |

Table 6: Relative errors of the network predictions of the MuJoCo environments on the test set, reported as avg±stdev over 4 runs.

### B.3 Node classification

To facilitate comparison with previous work, we follow the experimental methodology of [61].

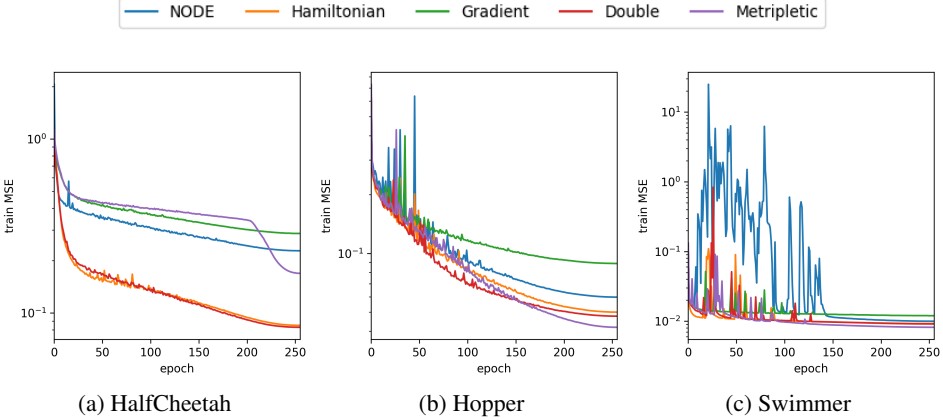

(a) HalfCheetah  (b) Hopper  (c) Swimmer

Figure 7: [Mujoco] Train MSE over epoch for all considered dynamics models. For the nodal feature, along with the position or the angle of the body part/joint, the node velocities are also considered.

### B.3.1 Experiment details

**Datasets.** We consider the three well-known citation networks, Cora [58], Citeseer [59], and Pubmed [60]; the proposed models are tested on the datasets with the original fixed Planetoid traing/test splits, as well as random train/test splits. In addition, we also consider the coauthor graph, CoauthorCS [61] and the Amazon co-purchasing graphs, Computer and Photo [62]. Table 7 provides some basic statistics about each dataset.

| Dataset | Cora | Citeseer | PubMed | CoauthorCS | Computer | Photo |
|---|---|---|---|---|---|---|
| Classes | 7 | 6 | 3 | 15 | 10 | 8 |
| Features | 1433 | 3703 | 500 | 6805 | 767 | 745 |
| Nodes | 2485 | 2120 | 19717 | 18333 | 13381 | 7487 |
| Edges | 5069 | 3679 | 44324 | 81894 | 245778 | 119043 |

Table 7: Dataset statistics.

**Hyperparameters** The bracket architectures employed for this task are identical to those in Section 4 except for that the right-hand side of the Hamiltonian, gradient, and double bracket networks is scaled by a learnable parameter $\text{Sigmoid}(\alpha) > 0$, and the matrix $\mathbf{A}_2 = \mathbf{I}$ is used as the inner product on 3-cliques. It is easy to verify that this does not affect the structural properties or conservation character of the networks. Nodal features $\mathbf{q}_i$ are specified by the datasets, and edge features $\mathbf{p}_\alpha = (d_0 \mathbf{q})_\alpha$ are taken as the combinatorial gradient of the nodal features. In order to determine good hyperparameter configurations for each bracket, a Bayesian search is conducted using Weights and Biases [76] for each bracket and each dataset using a random 80/10/10 train/valid/test split with random seed 123. The number of runs per bracket was 500 for CORA, CiteSeer, and PubMed, and 250 for CoauthorCS, Computer, and Photo. The hyperparameter configurations leading to the best validation accuracy are used when carrying out the experiments in Table 4 and Table 5.

Specifically, the hyperparameters that are optimized are as follows: initial learning rate (from 0.0005 to 0.05), number of training epochs (from 25 to 150), method of integration (rk4 or dopri5), integration time (from 1 to 5), latent dimension (from 10 to 150 in increments of 10), pre-attention mechanism $\tilde{a}$ (see below), positive function $f$ (either $\exp$ or $\text{Squareplus}$), number of pre-attention heads (from 1 to 15, c.f. Remark A.14), attention embedding dimension (from $1\times$ to $15\times$ the number of heads), weight decay rate (from 0 to 0.05), dropout/input dropout rates (from 0 to 0.8), and the MLP activation function for the metriplectic bracket (either $\text{relu}$, $\text{tanh}$, or $\text{squareplus}$). The pre-attention is chosen

from one of four choices, defined as follows:

$$\tilde{a}\left(\mathbf{q}_i, \mathbf{q}_j\right) = \frac{\left(\mathbf{W}_K \mathbf{q}_i\right)^\mathsf{T} \mathbf{W}_Q \mathbf{q}_j}{d} \qquad\qquad \text{scaled dot product,}$$

$$\tilde{a}\left(\mathbf{q}_i, \mathbf{q}_j\right) = \frac{\left(\mathbf{W}_K \mathbf{q}_i\right)^\mathsf{T} \mathbf{W}_Q \mathbf{q}_j}{\left|\mathbf{W}_K \mathbf{q}_i\right| \left|\mathbf{W}_Q \mathbf{q}_j\right|} \qquad\qquad \text{cosine similarity,}$$

$$\tilde{a}\left(\mathbf{q}_i, \mathbf{q}_j\right) = \frac{\left(\mathbf{W}_K \mathbf{q}_i - \overline{\mathbf{W}_K \mathbf{q}_i}\right)^\mathsf{T} \left(\mathbf{W}_Q \mathbf{q}_j - \overline{\mathbf{W}_Q \mathbf{q}_j}\right)}{\left|\mathbf{W}_K \mathbf{q}_i - \overline{\mathbf{W}_K \mathbf{q}_i}\right| \left|\mathbf{W}_Q \mathbf{q}_j - \overline{\mathbf{W}_Q \mathbf{q}_j}\right|}, \qquad \text{Pearson correlation}$$

$$\tilde{a}\left(\mathbf{q}_i, \mathbf{q}_j\right) = \left(\sigma_u \sigma_x\right)^2 \exp\left(-\frac{\left|\mathbf{W}_K \mathbf{u}_i - \mathbf{W}_Q \mathbf{u}_j\right|^2}{2\ell_u^2}\right) \exp\left(-\frac{\left|\mathbf{W}_K \mathbf{x}_i - \mathbf{W}_Q \mathbf{x}_j\right|^2}{2\ell_x^2}\right), \text{exponential kernel}$$

**Network architectures.** The architectures used for this experiment follow that of GRAND [49], consisting of the learnable affine encoder/decoder networks $\phi, \psi$ and learnable bracket-based dynamics in the latent space. However, recall that the bracket-based dynamics require edge features, which are manufactured as $\mathbf{p}_\alpha = (d_0 \mathbf{q})_\alpha$. In summary, the inference procedure is as follows:

$$\mathbf{q}(0) = \phi(\mathbf{q}) \qquad\qquad\qquad\qquad \text{(nodal feature encoding),}$$
$$\mathbf{p}(0) = d_0 \mathbf{q}(0) \qquad\qquad\qquad\qquad \text{(edge feature manufacturing),}$$
$$(\mathbf{q}(T), \mathbf{p}(T)) = (\mathbf{q}(0), \mathbf{p}(0)) + \int_0^T (\dot{\mathbf{q}}, \dot{\mathbf{p}})\, \mathrm{d}t, \quad \text{(latent dynamics)}$$
$$\tilde{\mathbf{q}} = \psi\left(\mathbf{q}(T)\right), \qquad\qquad\qquad \text{(nodal feature decoding)}$$
$$\mathbf{y} = c(\tilde{\mathbf{q}}). \qquad\qquad\qquad\qquad \text{(class prediction)}$$

Training is accomplished using the standard cross entropy

$$H\left(\mathbf{t}, \mathbf{y}\right) = \sum_{i=1}^{|\mathcal{V}|} \mathbf{t}_i^\mathsf{T} \log \mathbf{y}_i,$$

where $\mathbf{t}_i$ is the one-hot truth vector corresponding to the $i^{\text{th}}$ node. In the case of the metriplectic network, the networks $f_E, g_E, g_S$ are 2-layer MLPs with hidden dimension equal to the latent feature dimension and output dimension 1.

### B.3.2 Additional depth study

Here we report the results of a depth study on Cora with the Planetoid train/val/test split. Table 9 shows the train/test accuracy of the different bracket architectures under two different increased depth conditions, labeled Task 1 and Task 2, respectively. Task 1 refers to fixing the integration step-size at $\Delta t = 1$ and integrating to a variable final time $T$, while Task 2 instead fixes the final time $T$ to the value identified by the hyperparameter search (see Table 8) and instead varies the step-size $\Delta t$. Notice that both tasks involve repeatedly composing the trained network and hence simulate increasing depth, so that any negative effects of network construction such as oversmoothing, oversquashing, or vanishing/exploding gradients should appear in both cases. For more information, Table 10 provides a runtime comparison corresonding to the depth studies in Table 9.

Observe that every bracket-based architecture exhibits very stable performance in Task 2, where the final time is held fixed while the depth is increased. This suggests that our proposed networks are dynamically stable and effectively mitigate negative effects like oversmoothing which are brought by repeated composition and often seen in more standard GNNs. Interestingly, despite success on Task 2, only the gradient and metriplectic architectures perfectly maintain or improve their performance during the more adversarial Task 1 where the final time is increased with a fixed step-size. This suggests that, without strong diffusion, the advection experienced during conservative dynamics has the potential to radically change label classification over time, as information is moved through the feature domain in a loss-less fashion.

**Remark B.1.** *It is interesting that the architecture most known for oversmoothing (i.e., gradient) exhibits the most improved classification performance with increasing depth on Task 1. This is perhaps due to the fact that the gradient system decouples over nodes and edges, while the others*

*do not, meaning that the gradient network does not have the added challenge of learning a useful association between the manufactured edge feature information and the nodal labels. It remains to be seen if purely node-based bracket dynamics exhibit the same characteristics as the node-edge formulations presented here.*

| CORA networks | Trainable Parameters | Integration Time |
|---|---|---|
| **Hamiltonian** | 60723 | 1.49625 |
| **Gradient** | 160772 | 14.82404 |
| **Double Bracket** | 30718 | 5.36151 |
| **Metriplectic** | 104088 | 7.53107 |

Table 8: The integration time and number of trainable parameters corresponding to the best networks trained on CORA. Note that integration time can be considered as a surrogate for depth, since the temporal step-size of each network is fixed to 1.

| Depth study (acc) | 1 layer | 2 layers | 4 layers | 8 layers | 16 layers | 32 layers | 64 layers |
|---|---|---|---|---|---|---|---|
| **Hamiltonian** | $75.0 \pm 1.4$ | $77.9 \pm 1.0$ | $62.0 \pm 0.6$ | $38.8 \pm 1.0$ | $32.0 \pm 0.8$ | $25.4 \pm 0.5$ | $17.1 \pm 1.3$ |
| **Gradient** | $69.6 \pm 1.1$ | $72.7 \pm 1.0$ | $74.5 \pm 1.1$ | $77.7 \pm 1.0$ | $80.2 \pm 1.2$ | $81.4 \pm 1.0$ | $82.0 \pm 1.2$ |
| **Double Bracket** | $77.8 \pm 1.1$ | $81.2 \pm 0.8$ | $83.9 \pm 1.1$ | $83.8 \pm 1.0$ | $80.1 \pm 1.3$ | $58.9 \pm 1.0$ | $19.3 \pm 1.4$ |
| **Metriplectic** | $61.0 \pm 1.6$ | $62.4 \pm 1.9$ | $62.0 \pm 0.8$ | $61.9 \pm 1.3$ | $61.6 \pm 1.0$ | $60.6 \pm 1.3$ | $61.2 \pm 1.0$ |
| **Hamiltonian** | $77.2 \pm 0.8$ | $77.5 \pm 1.2$ | $77.4 \pm 1.2$ | $77.0 \pm 0.6$ | $78.0 \pm 0.7$ | $77.8 \pm 1.2$ | $77.4 \pm 0.7$ |
| **Gradient** | $79.9 \pm 1.4$ | $79.9 \pm 0.6$ | $79.6 \pm 0.6$ | $79.6 \pm 1.1$ | $80.4 \pm 1.1$ | $80.5 \pm 0.9$ | $79.8 \pm 1.1$ |
| **Double Bracket** | $84.2 \pm 0.9$ | $84.5 \pm 1.3$ | $84.1 \pm 0.5$ | $84.2 \pm 1.3$ | $84.1 \pm 0.8$ | $83.8 \pm 0.9$ | $84.2 \pm 1.0$ |
| **Metriplectic** | $61.7 \pm 1.4$ | $61.0 \pm 0.9$ | $61.0 \pm 1.0$ | $61.4 \pm 1.3$ | $61.6 \pm 1.4$ | $61.7 \pm 1.2$ | $61.9 \pm 1.1$ |

Table 9: Accuracy results of a depth study on CORA. Test accuracies reported as mean±stdev over 10 runs with random train/valid/test splits. Top/bottom groups correspond to tasks 1 and 2 of the study, respectively, where Task 1 uses a fixed step-size while Task 2 uses a fixed integration domain.

| Depth study (time) | 1 layer | 2 layers | 4 layers | 8 layers | 16 layers | 32 layers | 64 layers |
|---|---|---|---|---|---|---|---|
| **Hamiltonian** | $0.038 \pm 0.003$ | $0.060 \pm 0.006$ | $0.106 \pm 0.011$ | $0.191 \pm 0.035$ | $0.308 \pm 0.035$ | $0.497 \pm 0.045$ | $0.874 \pm 0.035$ |
| **Gradient** | $0.053 \pm 0.005$ | $0.091 \pm 0.011$ | $0.159 \pm 0.015$ | $0.273 \pm 0.023$ | $0.470 \pm 0.031$ | $0.809 \pm 0.037$ | $1.44 \pm 0.038$ |
| **Double Bracket** | $0.068 \pm 0.007$ | $0.125 \pm 0.010$ | $0.232 \pm 0.023$ | $0.434 \pm 0.039$ | $0.770 \pm 0.068$ | $1.36 \pm 0.098$ | $2.52 \pm 0.103$ |
| **Metriplectic** | $0.161 \pm 0.010$ | $0.305 \pm 0.011$ | $0.574 \pm 0.014$ | $1.10 \pm 0.012$ | $2.13 \pm 0.040$ | $4.11 \pm 0.075$ | $7.86 \pm 0.129$ |
| **Hamiltonian** | $0.052 \pm 0.009$ | $0.067 \pm 0.013$ | $0.114 \pm 0.021$ | $0.200 \pm 0.040$ | $0.350 \pm 0.054$ | $0.613 \pm 0.056$ | $1.17 \pm 0.084$ |
| **Gradient** | $0.365 \pm 0.020$ | $0.680 \pm 0.012$ | $1.26 \pm 0.022$ | $2.26 \pm 0.058$ | $4.22 \pm 0.069$ | $8.38 \pm 0.451$ | $19.4 \pm 0.152$ |
| **Double Bracket** | $0.241 \pm 0.036$ | $0.394 \pm 0.046$ | $0.731 \pm 0.057$ | $1.44 \pm 0.132$ | $2.98 \pm 0.286$ | $5.42 \pm 0.609$ | $9.44 \pm 0.487$ |
| **Metriplectic** | $1.05 \pm 0.051$ | $2.05 \pm 0.097$ | $3.87 \pm 0.100$ | $7.35 \pm 0.149$ | $14.1 \pm 0.332$ | $27.2 \pm 0.378$ | $53.8 \pm 0.370$ |

Table 10: Runtime results of a depth study on CORA. Wall clock times reported as mean±stdev over 10 runs with random train/valid/test splits. Top/bottom groups correspond to tasks 1 and 2 of the study, respectively, where Task 1 uses a fixed step-size while Task 2 uses a fixed integration domain.

