_2 \dots i_{k+1}} = \frac{a_{K-1,i_1 i_2 \dots i_K}}{\sum_{i_K, \dots, i_{K-k-1}} a_{K-1,i_1 i_2 \dots i_K}},$$

*for any $0 \le k \le K - 2$. However, application of the combinatorial codifferential $d_{k-1}^\mathsf{T}$ appearing in $d_{k-1}^*$ will necessarily symmetrize this quantity, so that the asymmetry behind the attention mechanism is lost in this formulation.*

To illustrate how this works more concretely, consider the extension $K = 3$ to 2-cliques, and let $\mathcal{N}(i,j) = \mathcal{N}(i) \cap \mathcal{N}(j)$. We have the tensors $\mathbf{A}_2 = (a_{2,ijk})$, $\mathbf{A}_1 = (a_{1,ij})$, and $\mathbf{A}_0 = (a_{0,i})$ defined by