# OpenReview forum: "Reversible and irreversible bracket-based dynamics for deep graph neural networks"
_NeurIPS.cc/2023/Conference — NeurIPS 2023 poster_

### Official Review · Reviewer_dWFZ · 2023-07-06

**Soundness:** 3 good
**Presentation:** 2 fair
**Contribution:** 3 good
**Rating:** 5
**Confidence:** 2

**Summary:**

This paper provides a unified framework inspired the bracket-based dynamical system to analysis the oversmoothing problem in GNN. The past work may leverage the opposite physics concept such as the reversible processes, irreversible process and therefore it is not clear how such concept help to design GNNs, while the framework in this paper gives a deeper understanding. Leveraging the data-driven exterior calculus, this paper constructs four novel architectures which span the both reversibility and irreversibility spectrum using geometric brackets as a means of parameterizing dynamics abstractly without empirically assuming a physical model. Interestingly, it can reinterpretate the message-passing  and GAT as the fluxes and conservation balances of physics simulator.  It also generalize the attention mechanism which extends the nodal feature to higher order cliques and provide a unified evaluation of dissipation.  Emperically, the author compare the architectures derived from this framework with classical GNN,e.g., GAT, GDE over several benchmarks.

**Strengths:**

1. This paper is well-written and offers valuable insights into the construction of graph neural networks. As far as I am aware, the idea presented in this paper is innovative. However, since I lack a background in physics, I would appreciate hearing the suggestions of other reviewers who are familiar with bracket-based dynamical systems.

2. The experimental result in damped double pendulum and MuJoCO dynamics looks good.

3. The author made the code available, promoting reproducibility.

**Weaknesses:**

1. This paper may be a little hard to understand for the reader without physics background.

2. It seems that the algorithm proposed by the author does not outperform the baselines in node classification problem.

**Questions:**

I am curious about the precise computational complexity of the architectures proposed by the author, specifically in the context of N nodes or N neighbors. It would be beneficial to have concrete values rather than just the order of complexity. Additionally, it would be valuable to understand the advantages of these architectures in comparison to traditional graph neural networks (GNNs).

Regarding the node classification experiment, the algorithm's performance is generally comparable to the baseline, although it does appear weaker in some cases. However, it showcases excellent performance in physical systems like the double pendulum and MuJoCo dynamics. Could this be attributed to the notion that physics-inspired frameworks are inherently more suitable for describing and modeling physical systems rather than social networks?


**Limitations:**

Yes

---

> ### Author Rebuttal · Authors · 2023-08-09
>
> Thank you for your thoughtful review.  We are happy that you have enjoyed our paper and that you appreciate the results of the physics-based simulations.  Below are responses to your specific concerns in the “Weaknesses” section:
>
> 1. *This paper may be a little hard to understand for the reader without physics background.*
>
> This is a fair point, as the physics underlying our architectures is reasonably involved.  Unfortunately, we are limited by space constraints in the main manuscript and cannot give a comprehensive explanation whenever concepts are introduced.  However, note that we have included the Appendices A.1 and A.2 which provide an introductory primer to the graph exterior calculus and bracket-based dynamical systems.  Additionally, we also plan to add some more motivating examples and literature references to these Appendices, as well as go carefully through the body to ensure that the meaning of our results is preserved even if the analysis cannot be precisely followed.  If you have additional suggestions regarding how to improve the readability of our paper, we are happy to incorporate them.
>
> 2. *It seems that the algorithm proposed by the author does not outperform the baselines in node classification problem.*
>
> This is true in most cases.  However, note that the goal of this work was not to develop an architecture which achieves state-of-the-art performance: this would require more significantly more hyperparameter tuning than we have done here, as well as additional empirical modifications in terms of feature encoding, graph rewiring, etc.  Instead, we aim to bring some clarity to the roles of reversibility and irreversibility in modern GNN architectures using common benchmarks from both graph analytics and physics-based modeling.  By connecting ideas such as gradient flows and graph attention to bracket-based dynamical systems, we have shown that architectures such as GAT and GRAND are not conservative or totally dissipative, and that this has non-obvious consequences on network performance.
>
> Regarding your questions, please see the following responses:
>
> *I am curious about the precise computational complexity of the architectures proposed by the author, specifically in the context of N nodes or N neighbors. It would be beneficial to have concrete values rather than just the order of complexity. Additionally, it would be valuable to understand the advantages of these architectures in comparison to traditional graph neural networks (GNNs).*
>
> The primary advantages of the architectures presented here over traditional GNNs are (1) explicit and automatic preservation of desirable physical properties such as the first and second laws of thermodynamics, and (2) straightforward incorporation of the graph attention mechanism directly into the differential operators governing network evolution.  By posing attentional GNNs as bracket-based dynamical systems, we make these networks more interpretable, conceptually simpler, and guaranteed to satisfy useful properties (e.g., dynamical stability) which follow from established theory.   This leads to better analysis of network properties and performance, as well as architectures which provably respect physical laws often found in training data.
>
> To address the question of computational complexity:  note that our architectures are predicated on the graph exterior calculus, which encodes derivatives in terms of signed incidence matrices d_0, d_1, etc.  These derivatives are always local on k-cliques, and our attention matrices A_0, A_1, etc. are always diagonal.  This means that operations on graph entities are always sparse, and the complexity will therefore be linear in the graph size.  On the other hand, it will also depend on the number of matrix multiplies called for by the evolution equations, the dimension of the features, and the time integrator itself, along with any need to, e.g., compute automatic derivatives during the forward pass.  Based on this information, it can be seen that the Hamiltonian bracket is relatively cheap, and the Metriplectic bracket is the most expensive.  A rough estimate of this complexity in terms of numerical runtime on CORA is given in the new Table 11.
>
> *Regarding the node classification experiment, the algorithm's performance is generally comparable to the baseline, although it does appear weaker in some cases. However, it showcases excellent performance in physical systems like the double pendulum and MuJoCo dynamics. Could this be attributed to the notion that physics-inspired frameworks are inherently more suitable for describing and modeling physical systems rather than social networks?*
>
> There may be some truth to this.  You are right that we do notice better results from structure-preservation on physics-based examples where there is a clear notion of energy/entropy, and the goal is to reproduce the trajectories of dynamical systems.  Conversely, it appears that graph classification networks can achieve comparable performance with many different types of structure-preserving and structure-agnostic architectures, perhaps indicating that the dynamics of the internal state in these networks are not critical to understanding the underlying classification problem.  With that said, we also notice remarkable stability of our structure-preserving architectures with increasing depth (see the attached Table 10), which is a notable advantage over standard GNN architectures even in this case.

---

> > ### Comment · Reviewer_dWFZ · 2023-08-16
> > **Thanks for the reply**
> >
> > I am glad to see that the authors plan to add some more motivating examples for the reader without physics background. I have no further questions.

---

### Official Review · Reviewer_7QNL · 2023-07-06

**Soundness:** 3 good
**Presentation:** 3 good
**Contribution:** 3 good
**Rating:** 6
**Confidence:** 2

**Summary:**

This work provides a comprehensive overview of graph attention networks (GATs), shedding light on their fundamental concepts and principles. Additionally, the authors introduce a set of novel GNN architectures that leverage structure-preserving bracket-based dynamical systems. By incorporating these systems into GNNs, the proposed architectures aim to enhance the modeling capabilities and performance of graph-based machine learning tasks. Overall, the paper presents both a broad perspective on GATs and innovative approaches to further advance the field.

**Strengths:**

They give a detailed theoretical analysis of graph attention networks with exterior calculus techniques. They also provide a generalized attention mechanism and propose several novel structure-preserving GNNs with satisfying performance.

**Weaknesses:**

The table writing is not clear. Sometimes orange result is better than blue result while sometime blue is better.

**Questions:**

I'd like to know whether your analysis can explain why "Gradient“ model performs worse on heterophily graphs like Computer and Photo. Which physics system is suitable for heterophily graphs and which is suitable for homophily graphs?

**Limitations:**

They have already listed their limitations.

---

> ### Author Rebuttal · Authors · 2023-08-09
>
> Thank you for your thoughtful review.  We are glad that you appreciate our analysis of graph attention networks in the context of the graph exterior calculus, as well as our interpretation of the attention mechanism as a learnable inner product on features.  Please see below for responses to your specific concerns:
>
> *The table writing is not clear. Sometimes orange result is better than blue result while sometime blue is better.*
>
> It is true that blue and orange do not always mean “best” and “second best” in our Tables.  In particular, we have used orange to represent the best result of the bracket-based architectures, and blue to represent the best result used for comparison.  Although this has been mentioned at the beginning of Section 5, we will reiterate this in the captions of our Tables since it was confusing.
>
> *I'd like to know whether your analysis can explain why "Gradient“ model performs worse on heterophily graphs like Computer and Photo. Which physics system is suitable for heterophily graphs and which is suitable for homophily graphs?*
>
> This is a good question.  In general, there is not an easy way to determine whether a particular structure-preserving architecture will be suitable for a particular degree of homophily or heterophily, as it has been shown (see, e.g., [*] below) that the ability of a network to distinguish between neighboring nodes relies crucially on the spectra of the learned weights.  On the other hand, it can be informally expected that architectures which are fully or partially conservative – such as the Hamiltonian, Double Bracket, and Metriplectic systems – will have an easier time learning on highly heterophilic data.  Indeed, this is what we observe on Computer and Photo in Table 5, where we see that the double bracket architecture is most performant.  This is because the totally dissipative Gradient architecture, along with many other architectures based on gradient flows, has a strong tendency to assign similar predictions to nodes in the same local neighborhood, which is colloquially known as “oversmoothing”.  While it is possible in theory for a totally dissipative architecture to correctly classify highly heterophilic data, the connection of dissipation/diffusion to graph convolution appears to make this difficult in practice.
>
> [*]  Francesco Di Giovanni, James Rowbottom, Benjamin P. Chamberlain, Thomas Markovich, and Michael M. Bronstein, “Understanding convolution on graphs via energies”, *arXiv:2206.10991*, 2023.

---

### Official Review · Reviewer_y9dF · 2023-07-06

**Soundness:** 3 good
**Presentation:** 3 good
**Contribution:** 3 good
**Rating:** 6
**Confidence:** 3

**Summary:**

The work proposes structure-preserving bracket-based dynamical systems to learn physical systems using GNNs. The authors proposed four formulations depending on completeness and character (conservative or dissipative). The models are demonstrated to be effective in physical system and node classification tasks.

**Strengths:**

* The method is constructed based on solid mathematics, and incorporating attention as a learnable inner product seems quite natural.
* The authors suggested various formalisms depending on completeness and character under the notion of the abstract bracket formulations, which offers a general viewpoint for physical systems.
* The method is quite adaptive because it is demonstrated to be effective in various tasks, including physical system and node classification.

**Weaknesses:**

* It is unclear to the reviewer where is the authors' contribution and where it is from prior works, particularly the mathematical part. The presentation could be more clear to distinguish existing work and the contribution of the work.
* The authors claim, "We use these architectures to systematically evaluate the role of (ir)reversibility in the performance of deep GNNs." However, the reviewer could not catch "the role of (ir)reversibility" from the experiments. The authors could elaborate on this point in the experiments part.
* Despite the broad range of experiments considered, the performance of the models is moderate. Therefore, it is unclear in which condition the proposed models are effective (see the following weakness for more details).
* In Section 5.1, the Hamiltonian model works better than the gradient model on the dissipative phenomena. The author could explain why the conservative works better here as it could be key to understanding "the role of (ir)reversibility."
* In addition, there are no experiments where the gradient model works best. With these results, it is unclear when the gradient model is effective. The authors could add experiments where the gradient works fine, e.g., incomplete and totally dissipative system.
* In Section 5.3, the authors reported that the proposed model consumed much memory. In this case, computation time for prediction could also be reported because there are typically tradeoffs between speed and accuracy.

Minor points:
* Pointers to "Table 3" (49, 124, and 159) might be "Table 1."

**Questions:**

* Does the method work well for PDEs, e.g., heat and Navier–Stokes equations?

**Limitations:**

The authors adequately addressed the limitations.

---

> ### Author Rebuttal · Authors · 2023-08-09
>
> Thank you for your thoughtful review. We are glad that you appreciate the idea of attention as a learnable inner product, and the versatility of our bracket-based architectures in capturing physical principles and performing a variety of tasks.
>
> We believe the primary weakness mentioned in your review – that it is not obvious where our mathematical contributions are relative to related works – can be cleared up easily. First, the work on data-driven exterior calculus in [10] deals fundamentally with identifying elliptic PDEs for data-driven physics modeling, and has nothing to do with dynamical systems, conservation properties, or bracket formalisms. Moreover, [10] presents no analysis related to graph attention or existing GNN architectures in terms of learnable inner products. So, while it is true that both [10] and our paper rely on the graph exterior calculus, their goals and contributions are essentially disjoint. Similarly, the work in [28] has a fundamentally different focus: they consider one specific type of bracket-based dynamical system, the metriplectic system, with the goal of learning physics outside the graph setting. Moreover, that work makes no use of the graph calculus or learnable inner products, meaning that [28] also has no direct connection to graph attention or existing GNN architectures.
>
> Responses to your remaining concerns are below. Please note truncation due to the character limit.
>
> *The authors claim, "We use these architectures to…*
>
> You are right that it is not easy to precisely characterize when one should use a reversible GNN architecture over an irreversible one. Our experiments show that both types of architectures can achieve competitive performance, though it seems that combining both conservative and dissipative dynamics is often the best choice. This suggests that the precise role of (ir)reversibility is context-dependent:  in physics-based modeling there are obvious reasons to use a network which is invariant-preserving, but enforcing physics may not be as critical for graph classification. Regarding depth, please see the attached Tables 9-11 and the global reviewer response.
>
> *Despite the broad range of experiments…*
>
> This is fair criticism, as our architectures only achieve the best performance in some of the experiments discussed. However, note that the goal of this work was not to develop an architecture which achieves state-of-the-art performance: this would require significantly more hyperparameter tuning, as well as additional ad hoc modifications in terms of feature encoding, graph rewiring, etc. Instead, we aim to establish a unified framework for understanding graph networks, in the process bringing some clarity to the roles of reversibility and irreversibility in modern GNN architectures using benchmarks from both graph analytics and physics-based modeling. By connecting ideas such as gradient flows and graph attention to bracket-based dynamical systems, we have shown that architectures such as GAT and GRAND are not conservative or totally dissipative, and that this has non-obvious consequences on network performance.
>
> *In Section 5.1, the Hamiltonian model works better…*
>
> This is a good point. The key to understanding this is to note that, in order to achieve coordinate-independence, our brackets are wrapped in a feature encoder/decoder.  This is necessary for learning bracket-based dynamical systems from arbitrary data: it is highly unlikely that a given set of data will match the form of our structure-preserving brackets in any arbitrary set of user-collected features, even if this data is drawn from a system which is physics-constrained. So, the autoencoder provides additional flexibility to our architectures, but comes with a challenge: the bracket structure we prescribe is enforced only in the latent space. Section 5.1 shows that it is possible for machine learned conservative dynamics to still produce an effective model for the dissipative double pendulum system when combined with a learnable encoding scheme.
>
> In simple problems, this issue is circumvented by using data with specially designed features which match a known bracket structure, e.g., position and momentum for canonical Hamiltonian systems. This removes the need for an autoencoder, but also limits the ability of bracket-based architectures to discover unknown physics or apply to less physical problems such as graph classification. For this reason, we have chosen to focus on the more general case of arbitrary features.
>
> *In addition, there are no experiments…*
>
> You are right that our experiments do not show a case where the gradient system performs the best. On the other hand, the original depth study in Table 8 shows that the gradient system appears to be the highly stable on node classification problems with respect to perturbations in the integration domain. See, e.g., Remark B.1.
>
> *In Section 5.3, the authors reported…*
>
> This is a good suggestion. We have included some computational timings on CORA in Table 11 as part of the new depth study. Note that the memory issues which were observed have been mitigated: they no longer appear in the new version of the code, though the metriplectic bracket is still prohibitively expensive on the large node classification problems despite its linear scaling in the graph size, due to its reliance on automatic differentiation during the forward pass. We anticipate that the metriplectic model will still have substantial impact in the context of data-driven physics models: in these settings feature dimensions are low. For evidence in this direction, please refer to the results in Tables 2 and 3.
>
> *Does the method work well for PDEs…?*
>
> Metriplectic dynamics have tremendous potential for multiphysics modeling. Reduced-order physics models obey a fluctuation-dissipation theorem, and the framework introduced here provides a means of learning non-equilibrium statistical mechanics. We are currently investigating this in other projects.

---

> > ### Comment · Reviewer_y9dF · 2023-08-16
> >
> > Thank you for the detailed responses, which addressed my concerns raised in the first review. Assuming these responses are incorporated in the revised manuscript, I have raised my scores (presentation: 2 -> 3, rating 4 -> 6). I generally appreciate the mathematical contribution and the broad range of experiments made by the work.
> >
> > > So, the autoencoder provides additional flexibility to our architectures, but comes with a challenge: the bracket structure we prescribe is enforced only in the latent space.
> >
> > This is an important point that could be clarified in the manuscript (e.g. limitation part) because most readers (including myself) expect this kind of model to respect physical laws in the original space in addition to the latent space through inductive biases.

---

> > > ### Author Response · Authors · 2023-08-16
> > >
> > > We are glad that we could assuage your concerns.  Thanks for being open to modifying the initial score.  We will make sure to explicitly clarify the mentioned limitation in future versions of the paper.

---

### Official Review · Reviewer_Ui2h · 2023-07-25

**Soundness:** 3 good
**Presentation:** 2 fair
**Contribution:** 3 good
**Rating:** 6
**Confidence:** 2

**Summary:**

This paper proposes a bracket-based dynamical system framework to design structure-preserving graph neural networks (GNNs). Specifically, the authors leverage four formalisms: Hamiltonian, Gradient, Double-Bracket, and Metriplectic, that model physical systems with different completeness and dissipation characteristics, and implement them via discretizations from exterior calculus. The authors parameterize GNNs with a chosen dynamic by specifying the bracket matrices and learning the energy/entropy functions via graph attention mechanisms. Experiments on real-world graph inference tasks show that structure-preserving GNNs can perform better than those that are not strictly structure-preserving, and the best dynamics depend on the downstream task.

**Strengths:**

Strengths:

1. It is novel to use bracket-based parameterizations to design physics-inspired GNNs.

2. It is fruitful to analyze existing GNNs (e.g., GAT, GRAND) under the bracket-based dynamical framework.

**Weaknesses:**

Weaknesses:

1. Since the paper is intended for general audience in graph machine learning, it can be written in a clearer manner with more motivating examples and targeting applications. For starters, include a notation section.

2. One of the key motivations of the paper is to improve performance of deep GNNs, yet all the experiments in the main manuscripts do not investigate explicitly the role of depth - are the GNNs used in these experiments actually deep? The more related experiments are described and discussed in Appendix B.3.2 as ablation study, but results (Table 8) show that for node classification task, only Gradient-based GNNs perform well when the number of layers increases, whereas other dynamics exhibit (significant) performance drop. If the paper intends to mitigate over-smoothing, the authors should provide more compelling experiments. On the other hand, if the paper simply intends to describe a physics-inspired framework to parameterize GNNs, then the authors should re-write the discussion of deep GNNs.



**Questions:**

1. What's the implications for practitioners? What applications would we expect imposing certain bracket structure outperforms the unconstrained ones? What structural requirement one should choose for different applications (e.g., add a column in Table 1 giving example applications)? The current set of experiments reads a bit confusing to me. For example, experiment 2 considers the node classification task, which requires some sense of clustering / smoothing, but not over-smoothing. So the findings in Table 4 showing that Double Bracket performs the best aligns with our expectations? But in Table 8, the Double Bracket formalism cannot be computed for deeper architectures - Is it merely a computational constraint? More explicit remarks and discussions will be great!

2. The role of (learnable inner product) $A$: the authors connect this with graph attention mechanism; but a priori one can also choose other forms of $A$? Have the authors compare some simple baselines, say fixed $A$ or even just set $A$ as identity? This could also help address the memory issue in experiments (Sec 5.3) where the authors cannot compute $A$ (and thus $E, S$) due to the high-dimensionality of the node features?

3. The authors discussed higher-order attention mechanism based on higher-order cliques (Appendix 4). Will this be useful in practice or is this a pure theoretical discussion? If the former, can the authors provide experimental demonstration? If the latter, the authors might consider removing this from the primary contributions since this does not seem like the focus of the paper?

**Limitations:**

This work does not have negative societal impact. The limitations are discussed in Section 6.

---

> ### Author Rebuttal · Authors · 2023-08-09
>
> Thank you for your thoughtful review.  We are happy to hear that you enjoyed our bracket-based parameterizations and that you appreciate the analysis of existing GNN architectures in terms of this framework.  Your weakness related to the clarity of the exposition has been well received: we plan to include a section at the beginning of Appendix A which summarizes our notation, and we will also add some discussion of additional examples and applications in Appendices A.1 and A.2, along with references where the interested reader can find more information.  Additionally, we have gone through the body of the paper again and made sure that the takeaways of our work are clear even if the analysis cannot be precisely followed.  If you have other suggestions, we are happy to incorporate them.
>
> To address your weakness related to the role of depth in our networks, we have modified the depth study as described in the global reviewer response, so that the role of depth in our architectures is more apparent.  Particularly, Tables 9-11 provide additional information related to the role of depth in the trained CORA models, and it can be seen from Table 10 that the performance of all bracket-based architectures does not depend critically on the depth in terms of number of layers, maintaining itself very well when the depth is increased while the interval of integration is held constant.
>
> To address your specific questions, please see below:
>
> 1) *What's the implications for practitioners? What applications would we expect imposing certain bracket structure outperforms the unconstrained ones? What structural requirement one should choose for different applications (e.g., add a column in Table 1 giving example applications)? The current set of experiments reads a bit confusing to me. For example, experiment 2 considers the node classification task, which requires some sense of clustering / smoothing, but not over-smoothing. So the findings in Table 4 showing that Double Bracket performs the best aligns with our expectations? But in Table 8, the Double Bracket formalism cannot be computed for deeper architectures - Is it merely a computational constraint? More explicit remarks and discussions will be great!*
>
> First, please note that the mentioned deficiency in the Double Bracket architecture was artificial and has been addressed since our submission (please see the global reviewer response for details).  As to the impact of our work for practitioners, this is a complex question.  We notice that, in almost every case, a combination of reversible and irreversible dynamics is necessary for optimal performance, which agrees with the intuition that network dynamics should simplify feature information without homogenizing it.  As for the question of when to choose structure-preserving networks over unconstrained ones: we chose not to focus on this at the present time, as there are many papers in the literature (e.g., [6, 19-26]) which have established the benefits of structure-preservation in machine learning – including dynamical stability, improved performance with depth, and relative insensitivity to initialization – on several different classes of problems.  Our work adds to this literature by showing that, perhaps counterintuitively, comparable performance can also be obtained on node classification experiments with architectures based on several different physical mechanisms.
>
> 2) *The role of (learnable inner product) A: the authors connect this with graph attention mechanism; but a priori one can also choose other forms of A? Have the authors compare some simple baselines, say fixed A or even just set A as identity? This could also help address the memory issue in experiments (Sec 5.3) where the authors cannot compute A (and thus E,S) due to the high-dimensionality of the node features?*
>
> It is certainly possible to choose other forms for the learnable matrix A which do not connect to the standard idea of graph attention: the only fundamental requirement A should have is positive definiteness to induce a valid norm.  For example, one could choose learnable diagonal matrices as in [10], or parameterize A in terms of learnable Cholesky factors.  We have indeed experimented with setting A to identity, but find that including an attentional A improves performance, likely for the same reasons that it is advantageous to incorporate graph attention in standard GNNs.  We have also experimented with setting the A matrices to update only at the start of every epoch, so that they are constant during the integration (this is referred to as “constant attention” in [48]).  We find that this choice makes very little difference in our experiments, so we have not included a comparative study to this effect.
>
> 3) *The authors discussed higher-order attention mechanism based on higher-order cliques (Appendix 4). Will this be useful in practice or is this a pure theoretical discussion? If the former, can the authors provide experimental demonstration? If the latter, the authors might consider removing this from the primary contributions since this does not seem like the focus of the paper?*
>
> While the discussion in the paper is purely theoretical, we expect higher-order attention to be useful on datasets with, e.g., explicit edge features or relatively dense connectivity.  We had difficulty identifying an appropriate edge feature dataset when we were designing the experiments for this work.  However, since the time of submission, we have improved the implementation of our attentional inner products, including the 2-clique attention which appears in the edge equation of the Gradient bracket.  These changes do appear to yield a modest performance increase, and this code will be released following the review process.

---

> > ### Comment · Reviewer_Ui2h · 2023-08-12
> > **Clarification of Table 9 and 10: interpretation of depth**
> >
> > Thank you for your detailed response! I have two follow-up questions below:
> >
> > Question:
> > 1. The new Table 9 caption reads "Note that integration time can be considered as a surrogate for depth, since the
> > temporal step-size of each network is fixed to 1". On the other hand, in the global response, the authors comment that "...but now progressively reduces the step-size of their 4th-order Runge-Kutta time integration from 1 to 1/64 ...". So Table 9 still uses a fixed step-size, while Table 10 uses a decreasing step-size?
> >
> > 2. If we adopt the interpretation of "integration time $\approx$ depth when step size is fixed to 1" in Table 9, then shouldn't the same analogy hold for Table 10, where the **"64" layers should effectively be much smaller**, since it should be a sum of $1 + 1/2 + 1/3 \ldots 1/64 \approx \ln 64 + \gamma \approx 4.7$ (i.e. the 64-th harmonic number)? If so, is the interpretation of "64" layer somewhat misleading, especially comparing to "standard GNNs, including GATs, are known to experience significant degradation after ~4 layers"?

---

> > > ### Author Response · Authors · 2023-08-14
> > >
> > > Thanks for your continued interest in our paper.  Please see our responses below for answers to your questions:
> > >
> > > 1.  *The new Table 9 caption reads "Note that integration time can be considered as a surrogate for depth, since the
> > > temporal step-size of each network is fixed to 1". On the other hand, in the global response, the authors comment that "...but now progressively reduces the step-size of their 4th-order Runge-Kutta time integration from 1 to 1/64 ...". So Table 9 still uses a fixed step-size, while Table 10 uses a decreasing step-size?*
> > >
> > > Not exactly.  The final time of integration for each bracket architecture is fixed according to the value in Table 9, and the step-size is constant for each entry in Table 10.  This means that, for each column in Table 10, we choose step_size = final_time / num_layers, so that time integration is performed at variable degrees of temporal resolution to test stability with increasing depth.
> > >
> > > 2. *If we adopt the interpretation of "integration time $\approx$ depth when step size is fixed to 1" in Table 9, then shouldn't the same analogy hold for Table 10, where the "64" layers should effectively be much smaller, since it should be a sum of $1 + 1/2 + 1/3 \ldots 1/64 \approx \ln 64 + \gamma \approx 4.7$ (i.e. the 64-th harmonic number)? If so, is the interpretation of "64" layer somewhat misleading, especially comparing to "standard GNNs, including GATs, are known to experience significant degradation after ~4 layers"?*
> > >
> > > Again, there is no step-size decrease during the inference procedure, and the meaning of depth is the same in both cases.  During network training, the step-size is fixed to 1.  Therefore, a final time of, e.g., 11, means that (in a forward Euler scheme) 11 network compositions are performed to reach the network’s predictions, which is why we say integration time is approximately depth in Table 9.  This interpretation continues in Table 10, where the step-size is changed to create a variable number of network compositions for the same final integration time:  from 2 up to 64.  This shows that the performance of the bracket-based architectures does not degrade with increasing depth, in contrast to more standard GNNs.
> > >
> > > Let us know if this answers your questions.  If not, we are happy to continue this discussion.

---

> > > > ### Comment · Reviewer_Ui2h · 2023-08-15
> > > > **Thank you and further clarifications**
> > > >
> > > > Thank you for your explanation. However, in the global response, you wrote
> > > > " This new study, which has been designed to supplement Table 8 in Appendix B.3.2, takes our bracket-based networks trained on CORA with Planetoid splits, but now progressively **reduces the step-size of their 4th-order Runge-Kutta time integration from 1 to 1/64** and tests their predictive accuracy on random splits."
> > > >
> > > > While you clarified above saying in Table 9-10, the step-size is constant. Do you refer to different "step-size" parameter in the global response versus your response here? Please clarify. Thanks!

---

> > > > > ### Author Response · Authors · 2023-08-15
> > > > >
> > > > > I understand the confusion now, thanks for clarifying. We perform a refinement study, showing results taking 2 steps, 4 steps, 8 steps...64 steps. For each experiment, the timestep size is 1/numbersteps.

---

> > > > > > ### Comment · Reviewer_Ui2h · 2023-08-18
> > > > > > **Clarification of refinement study**
> > > > > >
> > > > > > Thank you for the clarification. Is the refinement study the same as the "depth study" in Table 10, or a separate study?

---

> > > > > > > ### Author Response · Authors · 2023-08-20
> > > > > > >
> > > > > > > Yes, the reviewer's understanding is correct.

---

### Author Rebuttal · Authors · 2023-08-09

Thank you all for your insightful comments and useful feedback on our work.  We have heard the shared criticism that (1) parts of our manuscript could be difficult to understand for a general machine learning audience, and (2) that the role of depth in our architectures could be made clearer in the text.   We are taking steps to address these concerns.

Unfortunately, some technical complexity is unavoidable due to the mathematical machinery employed.  However, there are some additional things that we can do to ease the burden on our readers.  We have gone through the main manuscript and ensured that the practical implications of our work are clear, even if the analysis is difficult to follow.  Due to space constraints, it is difficult to include any more introductory details in the main body of the manuscript.  However, note that we have already written introductions to the graph exterior calculus in Appendix A.1 and to bracket-based dynamical systems in Appendix A.2, with the goal of making our manuscript accessible to the general machine learning community.  Following one reviewer’s suggestion, we have added a glossary of terms for ease of reference. We also plan to add more examples and common applications to these Appendices as per your suggestions.

Regarding the role of depth: we find that there is not an obvious relationship between depth and the performance of structure-preserving network architectures.  On the other hand, our experiments show that the bracket-based architectures presented here **can maintain their performance up to (and potentially beyond) 64 layers**, as demonstrated clearly in the new Table 10, while standard GNNs, including GATs, are known to experience significant degradation after ~4 layers.   In particular, many of the networks trained in Section 5 use the equivalent of 5-15 layers to reach their optimal predictions (see, e.g., the new Table 9).  Note that these new Tables correspond to a redesigned depth experiment which clarifies the influence that depth has on our architectures.  This new study, which has been designed to supplement Table 8 in Appendix B.3.2, takes our bracket-based networks trained on CORA with Planetoid splits, but now progressively reduces the step-size of their 4th-order Runge-Kutta time integration from 1 to 1/64 and tests their predictive accuracy on random splits.  This experiment tests the effect of increasing the depth of a network while holding its optimized domain of integration fixed and measures dynamical stability with increasing depth.  In the numerical analysis community, it is well known that the temporal step-size must be sufficiently small to achieve a stable result (the so called CFL conditions), which may not be respected by the experiment in Table 8 which fixes the temporal step-size to 1 and performs integration over progressively longer domains. The previous experiment (Table 8) was originally taken to match the methodology used in the GRAND paper, but we believe that Table 10 provides a more meaningful comparison.

In addition to these changes, we would like to reiterate that a primary contribution of our work is the principled reframing of graph neural networks in terms of bracket-based dynamical systems, which provides a rigorous way to both construct new network architectures as well as contextualize current ones.  Besides simplifying the analysis of existing architectures such as GAT and GRAND, this framework has great potential for data-driven physics applications, since reducing the order of a physics model naturally introduces dissipation in the form of information loss even when the full model is conservative.  Some evidence for the utility of the bracket formalism on physical problems is presented in Tables 2,3, and 6, and we are actively seeing success with this approach in other projects as well.

Finally, we would like to call your attention to the difference in results between the new depth study in Table 10 and Table 8 of the original submission.  This is because, in addition to reflecting a slightly modified experiment, the new results were obtained following a complete refactor of the code. Our original code was written around the publicly available GRAND codebase, but we had concerns related to memory usage and scalability of the GRAND code for deep networks (see missing data in tables 8 and 5). After reimplementing from scratch, we were able to remove the many hidden features of the GRAND code and avoid these issues. We are repeating all experiments with the new code but have completed preliminary tests that demonstrate the issues have been resolved, and this code will be made available following the review process.

---

### Decision · Program_Chairs · 2023-09-21

**Decision:**

Accept (poster)

**Comment:**

This paper proposes novel GNN architectures based on structure-preserving bracket-based dynamical systems. The reviewers appreciate the novel and solid contributions, and are generally positive about the paper. Most questions from the initial reviews were satisfactorily addressed. The authors committed to making the paper more accessible to a broader audience. We suggest the authors to incorporate the reviewers' suggestions in the final version.